# SEISMIC HAZARD MAPS OF PESHAWAR DISTRICT FOR VARIOUS RETURN PERIODS

Khalid Mahmood[1], Naveed Ahmad[2,*], Usman Khan[1], Qaiser Iqbal[1]

[1]Sarhad University of Science and Information Technology, Peshawar, KP Pakistan.
[2]Department of Civil Engineering, UET Peshawar, KP Pakistan.
[*]Correspondence E-Mail: naveed.ahmad@uetpeshawar.edu.pk

**Abstract:**

Probabilistic seismic hazard analysis of Peshawar District has been performed for a grid size of $0.01^0$. The seismic sources for the target location are defined as the area polygon with uniform seismicity. The earthquake catalogue was developed based on the earthquake data obtained from different worldwide seismological networks and historical records. The earthquake events obtained in different magnitude scale were converted into moment magnitude using indigenous catalogue-specific regression relationships. The homogenized catalogue was subdivided into shallow crustal and deep subduction zone earthquake events. The seismic source parameters were obtained using the bounded Gutenberg-Richter recurrence law. Seismic hazard maps were prepared for peak horizontal acceleration at bedrock using different ground motion attenuation relationships. The study revealed, the selection of an appropriate ground motion prediction equation is crucial in defining the seismic hazard of Peshawar District. The inclusion of deep subduction earthquakes does not add significantly to the seismic hazard for design base ground motions. The seismic hazard map developed for shallow crustal earthquakes, including also the epistemic uncertainty, was in close agreement with the map given in the Building Code of Pakistan – Seismic Provision (2007) for a return period of 475 years on bedrock. The seismic hazard maps for other return periods i.e. 50, 100, 250 475, and 2500 years are also presented.

**Keywords:** Seismic hazard map, probabilistic seismic hazard analysis, BCP-SP 2007, Peshawar, CRISIS

## 1. Introduction:

Peshawar is the capital city of the Khyber Pakhtunkhwa province of Pakistan that has historical background in the history of indo-subcontinent. The city provides key access to the Central Asian States

through Afghanistan along the Western borders of Pakistan. It is located at 710, 43.4' N latitude and 330, 93.7' E Longitude in the western Himalayan region.

Peshawar is characterized by high seismicity rates due to its proximity to the active plate boundary between the Indian and Eurasian plates, which are converging at the rate of 37-42 mm/year (Chen et al., 2000). The Main Boundary Thrust (MBT) system along which the devastating Kashmir earthquake occurred in 2005, is located in the northern parts of the country together with some other active regional fault systems; including Main Mantle Thrust (MMT) and Main Karakorum Thrust (MKT). These faults, if reactivated can act as a potential source of seismic hazard for the region including Peshawar (Waseem et al., 2013). This was confirmed also by the recent 2015 Afghanistan-Pakistan earthquake that caused widespread damages in the province of Khyber Pakhtunkhwa (Ahmad, 2015), including Peshawar, damaging a number of important structures in the historic city (**Fig. 1**).

The Building Code of Pakistan in 1986 has placed Peshawar in Zone 2 that corresponds to intensity V-VI on the Modified Mercalli Intensity scale. Mona Lisa et al. (2007), based on the probabilistic seismic hazard analysis for the NW Himalayan thrust, recommended a value of 0.15g for Peshawar. Hashash et al. (2012), using discrete faults model for Northern Pakistan, suggested peak ground acceleration (PGA) value in the range of 0.20-0.4g. Rafi et al. (2012), based on the probabilistic seismic hazard analysis and zonation for Pakistan and Azad Jammu and Kashmir, has evaluated a value of 0.175g for Peshawar. Several researchers either regionally or partially have studied the seismic hazard of Peshawar District (Table 1). The Geological Survey of Pakistan (2006) Seismic Zoning Map suggests a PGA value in the range of 0.03-0.1g, Zaman and Warnitchai (2010) suggests in the range of 0.33-0.40g while Zhang et al. (1999) suggested in the range of 0.166-0.244g. The Building Code of Pakistan – Seismic Provision (2007), which is a legal binding for the seismic design of structures in Pakistan, has placed Peshawar in Zone 2B. This zone has peak ground acceleration in the range of 0.16g to 0.24g for a return period of 475 years. This has revealed that previous seismic hazard studies of Peshawar and Northern Pakistan report widely conflicting results (Ahmad et al., 2019; Ambraseys et al., 2005; Khaliq et al., 2019; Sesetyan et al., 2018; Waseem et al., 2018, 2020).

The present study aims to re-calculate the seismic hazard of Peshawar, based on the up-to-date earthquake catalogue and ground motion prediction equations, and compare the same with that recommended by BCP-SP (2007).  The PGA value at bedrock was calculated using the classical probabilistic seismic hazard analysis procedure. The area sources as suggested by BCP-SP (2007) were for which the earthquake catalogue was obtained from worldwide seismogram networks and historical records. The Modified Gutenberg-Richter empirical model was used to calculate the seismic zone parameters for both shallow crustal and deep subduction zone earthquakes. The seismic hazard in terms

of PGA at bedrock was calculated and plotted in GIS tool. Different ground motion attenuation relationships compatible to the geology and seismicity of local environment were used to quantify model-to-variability in seismic hazard of Peshawar District. Furthermore, the logic tree approach was used to take into consideration the epistemic uncertainty. The GIS based seismic hazard map developed for a return period of 475 years was compared with that given in the BCP-SP (2007). Seismic hazard maps were prepared for various other return periods i.e. 50, 100, 250, 475 and 2500 years.

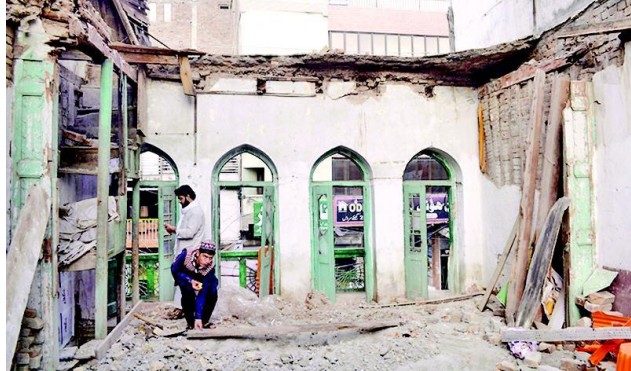

(a) Peshawar City, Qisa Khwani Bazar: Complete collapse of building roof.

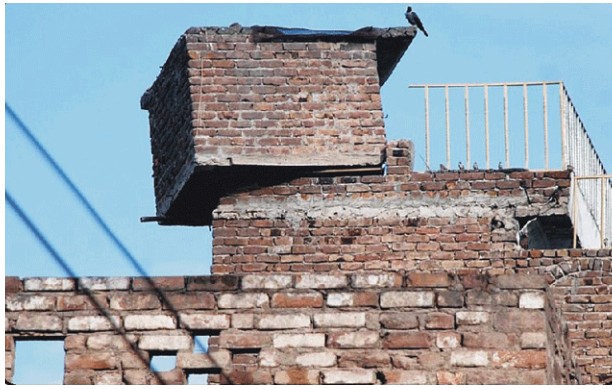

(b) Peshawar City, Ganj: Sliding of overhead tank on the building roof.

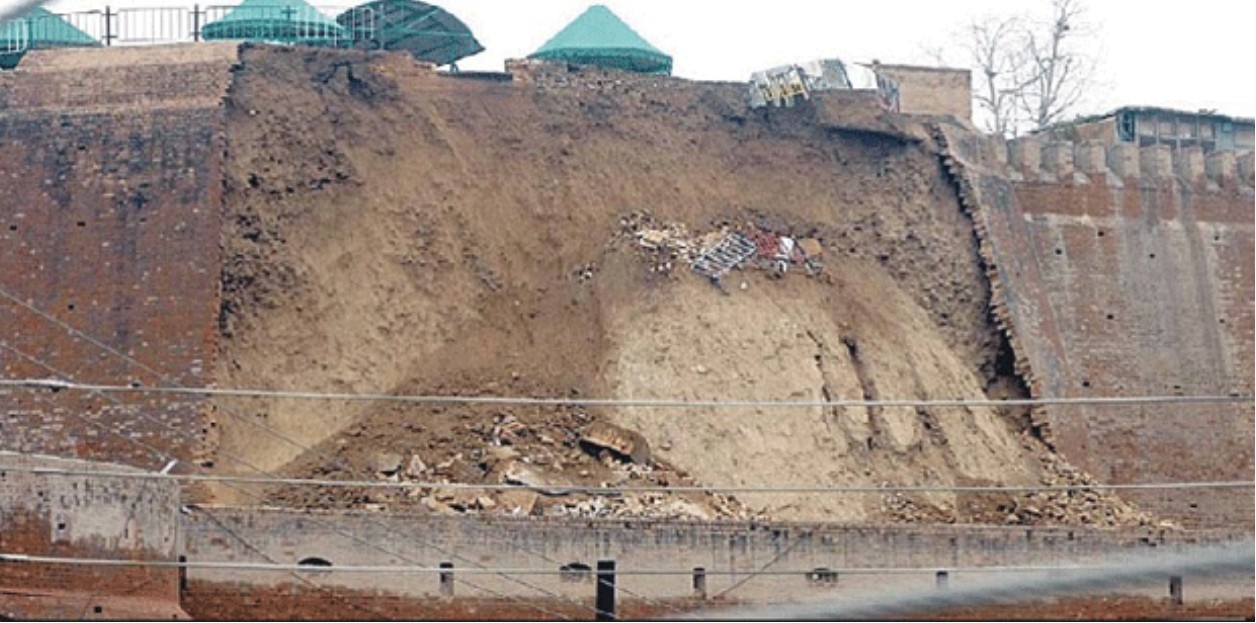

(c) Peshawar City, Fort: Collapse of masonry retaining wall and backfill sliding of Fort "Qilla Bala Hisar".

**Figure. 1** Damages observed in Peshawar during 2015 Afghanistan-Pakistan earthquake.

**Table 1.** Seismic hazard of Peshawar reported by various researchers

| S. No. | Authors | PGA (g) |
|--------|---------|---------|
| 1 | Bhatia et al. (1999) | 0.10 – 0.15 |
| 2 | Mona Lisa et al. (2007) | 0.15 |
| 3 | Zhang et al. (1999) | 0.16 – 0.24 |
| 4 | Rafi et al. (2012) | 0.17 |
| 5 | Hashash et al. (2012) | 0.20 – 0.40 |
| 6 | Şeşetyan et al. (2018) | 0.30 – 0.40 |
| 7 | Khaliq et al. (2019) | 0.32 – 0.34 |
| 8 | Zaman and Warnitchai (2012) | 0.33 – 0.40 |
| 9 | Waseem et al. (2020) | 0.33 |
| 10 | Waseem et al. (2018) | 0.38 |
| 11 | Shah et al. (2019) | 0.06 |
| 12 | Ahmad et al. (2019) | 0.16 to 0.24 |

## 2. Probabilistic Seismic Hazard Analysis

The uncertainties in the location, size and rate of recurrence of earthquake along with the variation in the ground motion intensity and spatial variability can be well considered in the probabilistic seismic hazard analysis procedures (Ornthammarath et al., 2011; Çağnan and Akkar, 2018; Rowshandel, 2018). The probabilistic seismic hazard analysis (PSHA) provides a framework in which these uncertainties can be identified, quantified, and combined in a rational manner to provide a holistic view of the seismic hazard.

According to the modified Gutenberg-Richter Law the earthquake exceedance rate $\lambda(M)$ for an earthquake magnitude $M$ can be defined using Equation (1);

$$\lambda(M) = \lambda_o \frac{e^{-\beta M} - e^{-\beta M_u}}{e^{-\beta M_0} - e^{-\beta M_u}} , M_o \leq M \leq M_u \qquad (1),$$

$\lambda_0$ is the exceedance rate in the range of lower $M_o$ and upper limit $M_u$ of magnitude, $\beta$ is the earthquake source parameter. Considering earthquake as a Poisson process, the probability density of the earthquake magnitude can be obtained using Equation (2):

$$P(M) = \lambda_o \beta \frac{e^{-\beta M}}{e^{-\beta M_o} - e^{-\beta M_u}} \qquad (2)$$

The strong ground motion parameters i.e. acceleration, velocity and displacement, are characterized using attenuation relationships that shows the variation in strong motion amplitude with source-to-site distance and depends on a number of source, path and site parameters (Douglas, 2019; Kramer, 1996; McGuire, 2004; Rupakhety and Sigbjörnsson, 2009). For example, the attenuation relationship for the peak horizontal acceleration has been developed by Campbell (1981) within 50 kM of fault rupture in magnitude 5.0 to 7.7 earthquake. Campbell and Bozorgnia (1994) developed attenuation relationships using worldwide moment magnitude in the range of 4.7 to 8.1. This relationship is more specific and provides additional terms for source characterization. Toro et al., (1994) has developed attenuation relation in term of peak horizontal acceleration on rock side for the continental portion of Northern America. Among others Boore and Atkinson (2008) and Akkar and Bommer (2010) have developed site specific attenuation relationship that can calculate peak acceleration in term of earthquake magnitude, source to site distance, fault mechanism and site condition. Boore and Atkinson (2008) model was developed based on the empirical regression of PEER NGA strong-motion database while that of Akkar and Bommer (2010) model was developed for Europe, Mediterranean and the Middle East region.

The mentioned attenuation relationships can be used for ground motion prediction of shallow crustal earthquakes. However, several researchers including Crouse et al. (1988), Crouse (1991), Molas and Yamazaki (1993), Youngs et al. (1995) have pointed out different conditions of attenuation relationships for shallow and subduction zones. Lin and Lee (2008) and Kanno et al. (2006) have developed attenuation relationships for earthquake records of Taiwan and Japan respectively. The study of Lin and Lee (2008) showed lower attenuation for subduction zones than that for crustal shallow earthquakes. Therefore, the use of shallow crustal earthquake attenuation relationships may lead to underestimation of the seismic hazard for subduction earthquakes in probabilistic analysis.

In probabilistic seismic hazard analysis, the peak acceleration at a location is a function of magnitude and distance that is lognormally distributed with standard deviation. In the hazard analysis, the study area is first divided into seismic sources based on tectonics and geotechnical characteristics. The different seismic sources are assumed to occur independently, and the seismic events are considered to occur uniformly over the source. The acceleration exceedance rates $v_i(a)$ for the single seismic source $i^{th}$ is calculated using Equation (3):

$$v_i(a) = \sum_i w_{ij} \int_{M_o}^{M_u} \left(-\frac{d\lambda_i(M)}{dM}\right) Pr(A > a|M, R_{ij})dM \qquad (3)$$

where $M_0$ is the smallest and $M_u$ is the largest magnitude of seismic source, $Pr(A>a|M,R_{ij}))$ is the probability that acceleration A exceeds the value a at distance $R_{ij}$ for an earthquake of magnitude M. The acceleration exceedance $v(a)$ due to all sources-N is calculated through combining all sources, as given in Equation (4):

$$v(a) = \sum_{i=1}^{N} v_i(a) \qquad\qquad (4)$$

## 3. Seismicity of Peshawar

The collision of Eurasian and Indian plate has resulted in the formation of active Himalayan orogenic system that is further classified into Tethyan, Higher, Sub and Lesser Himalayas (Gansser, 1964). The divisions are based on the tectonic blocks formed and separated by major faults boundary.

The Microsoft Encarta Reference Library (2003) shows that the valley of Peshawar, consists of southern part of Eurasian plate and northern part of Indo-Australian plate. This part of the Himalayas is variably interpreted to be Lesser Himalayas (Tahirkheli et al., 1982) and Tethyan Himalayas (DiPietro and Pogue 2004). The seismic hazard study of Waseem et al. (2006) has identified about twenty-one seismogenic faults around Peshawar. Most of these faults have reverse fault mechanism and have a Joyner-Boore distance $R_{JB}$ in the range of 19-100 km. According to Ali and Khan (2004), most of the significant earthquakes felt at Peshawar have their origin in the Hindu Kush region of Afghanistan and few in northern areas of Pakistan.

## 4. Case Study PSHA of Peshawar

The seismic hazard software CRISIS-2007 was used to calculate the peak acceleration at bedrock for Peshawar District. **Fig. 2** shows the geographical location of Peshawar District within the geo-political boundaries of KP Province of Pakistan. The hazard analysis requires seismic source geometry, earthquake reoccurrence relationship and the selected ground motion attenuation relationship. In the present study the ground motion attenuation relationships of Boore and Atkinson (2008) and Akkar and Boomer (2010) were used for shallow crustal seismic earthquakes and that of Lin and Lee (2008) and Kanno et al. (2006) for deep subduction zone earthquakes. The earthquake events within 50 km depth were considered as shallow while earthquake events occurring at depth larger than 50 km were considered as deep

1  earthquakes. The seismic hazard maps were prepared in GIS environment based on a grid size of $0.01^0$ for

2  various return periods i.e. 50, 100, 250, 475 and 2500 years.

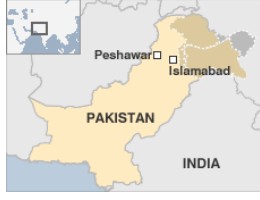
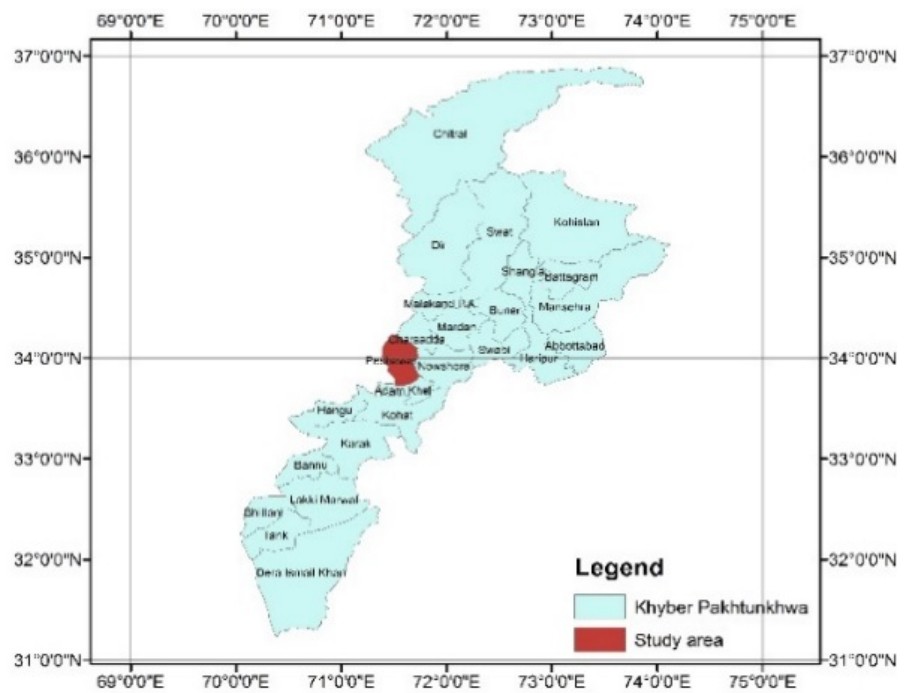

**Figure. 2** Location of study area

**4.1 Seismic Sources Identification and Characterization**

The Building Code of Pakistan (BCP-SP, 2007) has defined the potential shallow seismic sources for

Pakistan including northern areas. Those within 200 km of Peshawar were considered potential sources

for earthquake activity impacting Peshawar (**Fig. 3)**. The potential seismic sources (seven seismic sources

in present study) for Peshawar region in a rectangular shape with latitude (31.888~ 36.006) and longitude

(69.562~73.620) as shown in **Fig. 4,** were considered for the compilation of earthquake catalogue. The

earthquake catalogues were obtained using worldwide seismogram network sources i.e., United States of

Geological Survey, (USGS), National Geophysical Data Center (NGDC), Global Centroid Moment

Tensor (GCMT) and International Seismological Center (ISC) using the time span of 1500 AD till 2015

with focal depth up to 1000 m. The catalogue also included historical data from Ambrasey (2000) and

Ambrasey and Douglas (2004). Khan et al. (2008) also reported updated earthquake catalogue for

Pakistan, however, majority of their events relevant for Peshawar were already included in the catalogue

of present study for seismic sources characterization. These different networks already discussed gives

earthquake magnitude in different scales e.g. moment magnitude, surface magnitude and low magnitude,

etc. According to Kanamori (1977) and Hanks and Kanamori (1979) the moment magnitude is the most

accurate scale that does not saturate in higher magnitude events. Therefore, all the magnitudes were
converted into moment magnitude (**Mw**) using regression analysis. **Fig. 5** shows the empirical
relationships established in the present study based on the catalogue obtained for earthquake magnitude
conversion. These were used for the catalogue homogenization.

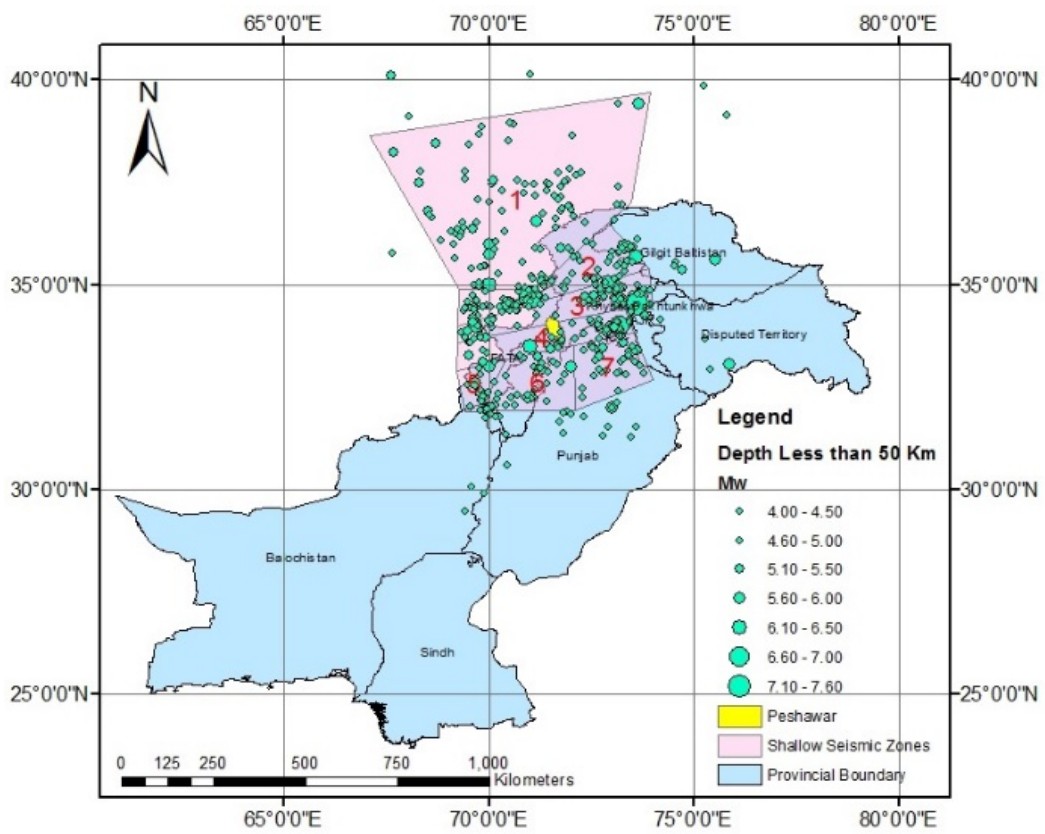

**Figure. 3** Shallow seismic sources for Peshawar (BCP-2007)

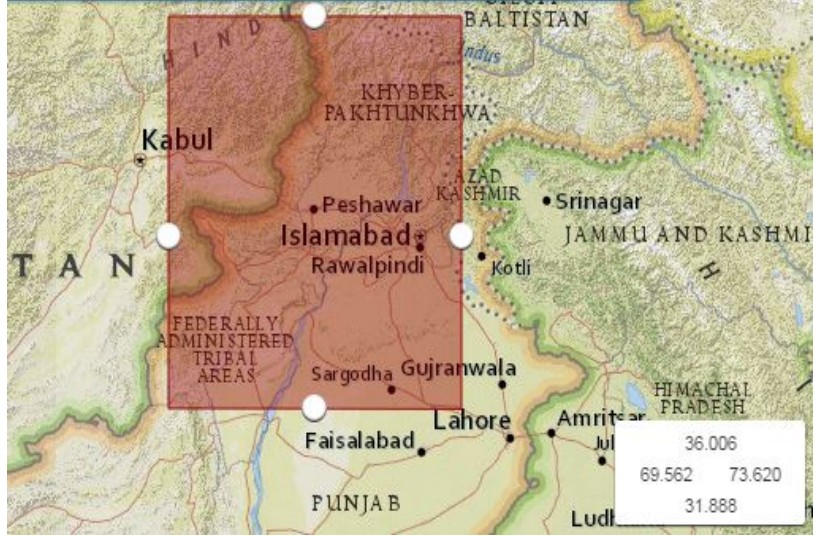

9  **Figure. 4** Seismic source identification with defined latitude and longitude. © Google Map.

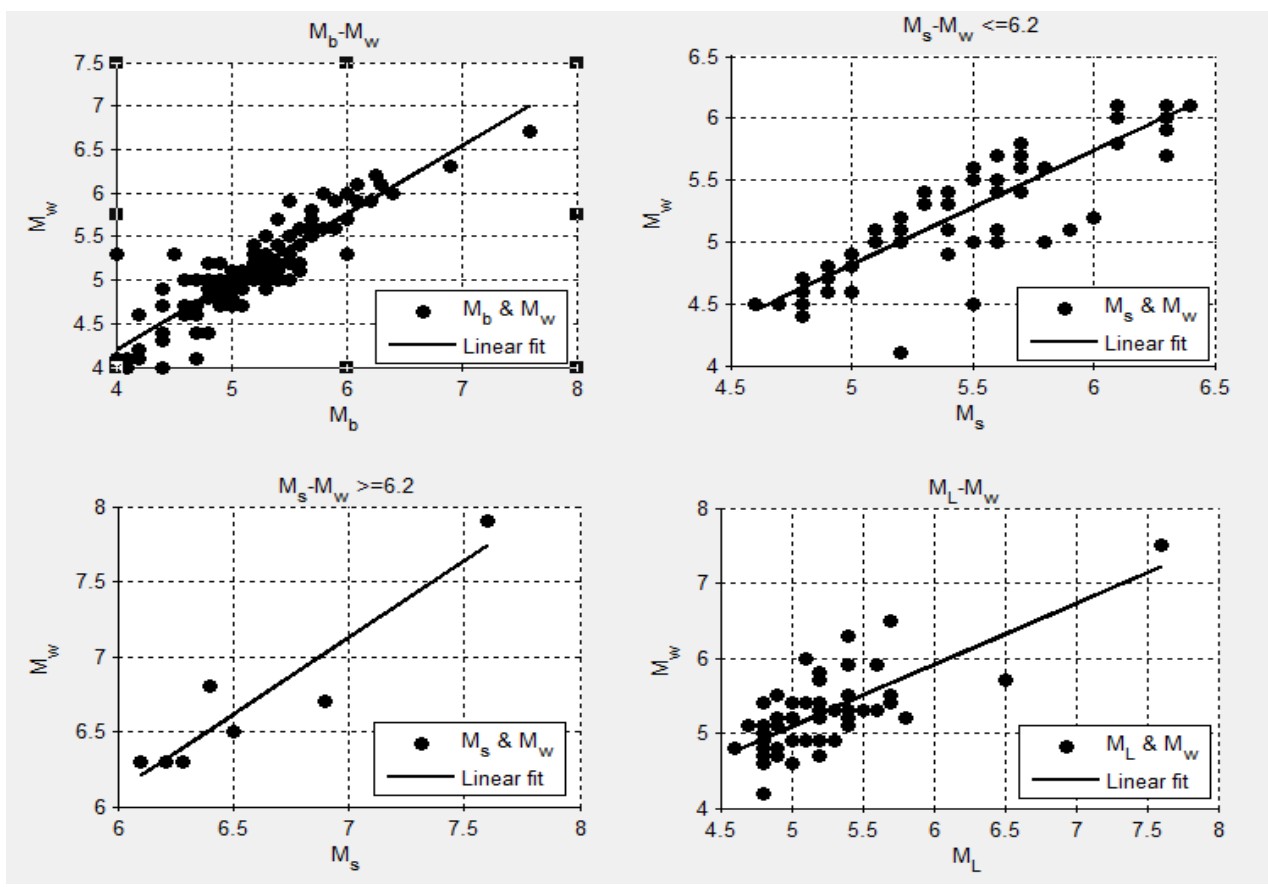

**Figure. 5** Empirical relationships for moment magnitude

The homogenized catalogue was further subdivided into shallow (depth less than 50 km) and deep (depth more than 50 km) earthquake events. **Fig. 6** shows the shallow and deep earthquake records along with seismic zones as defined in BCP-SP (2007). Furthermore, **Table 2** reports the number of earthquakes in each seismic source along with maximum and minimum magnitude of each source. Deep earthquakes were found primarily in seismic source 1 and seismic source 2 that included the Hindu Kush seismic region. The deep sources were selected in consultation with the National Center of Excellence in Geology, Peshawar. Since, deep sources were not studied before for Peshawar.

**Table. 2**: No. of earthquakes, minimum and maximum magnitude in shallow and deep seismic source

| Zones | 1 | | 2 | | 3 | 4 | 5 | 6 | 7 |
|---|---|---|---|---|---|---|---|---|---|
| Depth, (kM) | <50 | >50 | <50 | >50 | <50 | <50 | <50 | <50 | <50 |
| No. of Earthquakes | 99 | 454 | 79 | 23 | 76 | 43 | 17 | 35 | 32 |
| Minimum (Mw) | 4.0 | 4.0 | 4.1 | 4.2 | 7.6 | 4.0 | 4.1 | 4.1 | 4.1 |
| Maximum (Mw) | 6.2 | 7.7 | 4.0 | 5.1 | 7.5 | 6.8 | 6.0 | 6.0 | 5.5 |

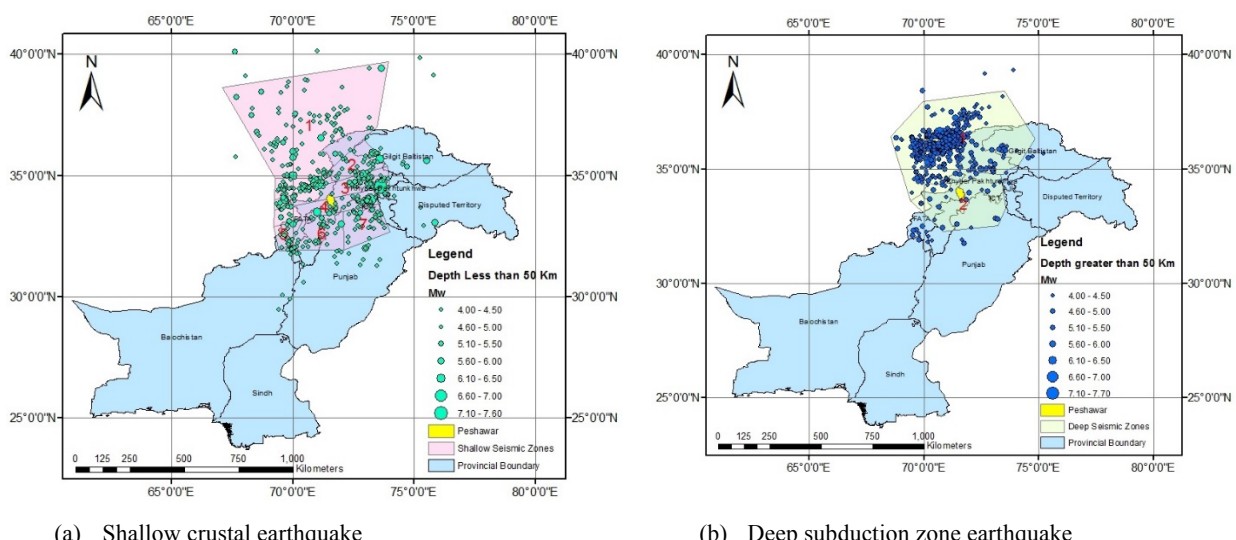

    (a)   Shallow crustal earthquake                     (b)   Deep subduction zone earthquake

**Figure. 6** Earthquake records from homogenized catalogue and with defined seismic sources

**4.2 Processing of Earthquake Catalogue**

*De-clustering*

In seismic hazard analysis the probability of earthquake occurrence is considered to follow a Poison's

process, which considers the independent events occurs randomly in time and space. Only mainly shocks

are considered for hazard analysis. This is to avoid over estimation of the seismic hazard. The dependent

events (foreshocks and aftershocks) are temporally and spatially dependent on the main shocks. For this

purpose Declustering was performed to remove the dependent events for the catalogue. The Gardner and

Kenopoff (1974) Declustering algorithm method was used for removing foreshocks and aftershocks

(Gardner and Knopoff, 1974). This performs windowing procedure in time and space on the event

magnitude to identify the dependent events. To perform theses analysis Z-Map coding developed by ETH

in Zurich (freely available) was used. The homogenized catalogue was converted into Z-map specified

format to perform the routine analysis. A total of 926 independent events remained after Declustering.

*Completeness Analysis:*

The catalogue also report events from very past, which cannot be considered complete for all the

magnitudes and time span. The time window starts from the year 1500, however, since then the catalogue

is not reported on regular basis. The instrumental observation of seismic data started after 1960, which

now observes and document complete details of the earthquake events on regular basis. Due to these

reasons the specified time window (1500-2015) cannot be considered in obtaining the activity rate, as this will result in underestimation of the activity rate. For this purpose completeness analysis was performed using visual cumulative method (CUVI) proposed by Mulargia and Tinti (1985). It is a simple procedure based on the observation that earthquakes follow a stationery occurrence process. It is used to find the completion point (CP) after which the catalogue is considered to be complete (Tinti and Mulargia, 1985). The procedure is to divide the magnitudes form 4 to 8 into various bands having 0.5 step-size. The selected bands are: 4.00 to 4.50. 4.51 to 5.00, 5.01 to 5.50, 5.51 to 6.00, 6.01 to 6.50, 6.51 to 7.00, 7.01 to 7.7. In each band cumulative the numbers of total earthquakes are plotted against the year of earthquakes, the period of completeness (Tc) is considered to begin at the earliest time when the slope of the fitting curve can be well approximated by a straight line (**Fig. 7**). **Table 3** reports the completeness points and time periods for each magnitude band.

**Table 3**:Completeness intervals and completion period of each magnitude band

| Magnitude Range | Average Magnitude | Completion Interval | Completion Period (Tc) |
|---|---|---|---|
| 4.00 – 4.50 | 4.25 | 1995 – 2015 | 20 |
| 4.51 – 5.00 | 4.75 | 1985 – 2015 | 30 |
| 5.01 – 5.50 | 5.25 | 1972 – 2015 | 43 |
| 5.51 – 6.00 | 5.75 | 1954 – 2015 | 61 |
| 6.01 – 6.50 | 6.25 | 1928 – 2015 | 87 |
| 6.51 – 7.00 | 6.75 | 1878 – 2015 | 137 |
| 7.01 – 7.77 | 7.35 | 1842 – 2015 | 173 |

## 4.3 Seismic Source Parameters

The modified Gutenberg-Richter reoccurrence law, as mentioned earlier, was used in the present seismic hazard analysis to characterize the G-R parameters. The seismic source parameters (i.e., $\eta_o$, $\beta$) were calculated from setting linear trend line to the graph between $log_{\lambda_m} \sim M_w$ as shown in **Fig. 8** for both shallow and deep earthquakes for all seismic zones. Table 4 reports the seismic sources' G-R parameters for all seismic sources and both shallow and deep earthquakes.

**Table. 4** Seismic source parameters for shallow and deep sources

| Seismic Source | 1 | 2 | 3 | 4 | 5 | 6 | 7 | 1* | 2* |
|---|---|---|---|---|---|---|---|---|---|
| $\lambda_o$ | 10.055 | 3.625 | 4.075 | 2.876 | 1.059 | 2.143 | 2.731 | 24.143 | 5.652 |
| $\beta = 2.303b$ | 2.832 | 2.03 | 2.10 | 2.24 | 2.03 | 2.10 | 2.5 | 2.17 | 2.97 |
| $M_u$ | 6.2 | 7.6 | 7.5 | 6.8 | 6.0 | 6.0 | 5.5 | 7.7 | 6.0 |

1,2,3,4,5,6 and 7 are shallow seismic sources and 1*, 2* are deep seismic sources

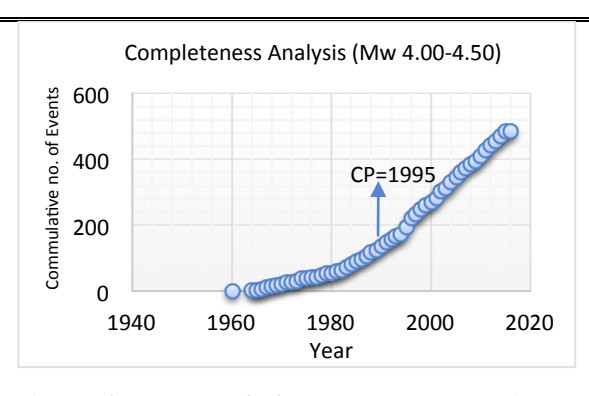

(Completeness period 20years:1995-2015)

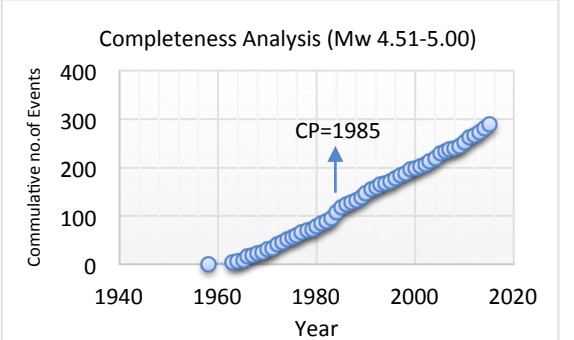

(Completeness period 30years:1985-2015)

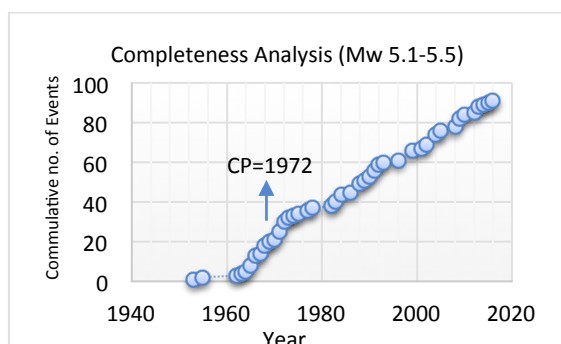

(Completeness period 43 years: 1972-2015)

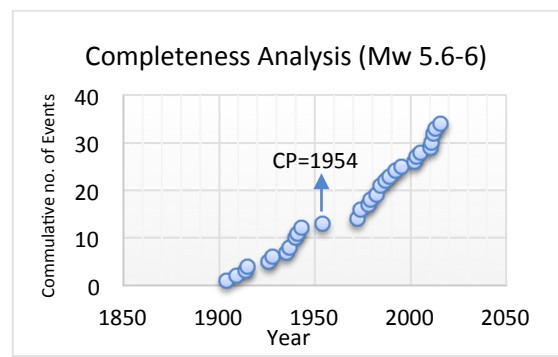

(Completeness period 61 years: 1954-2015)

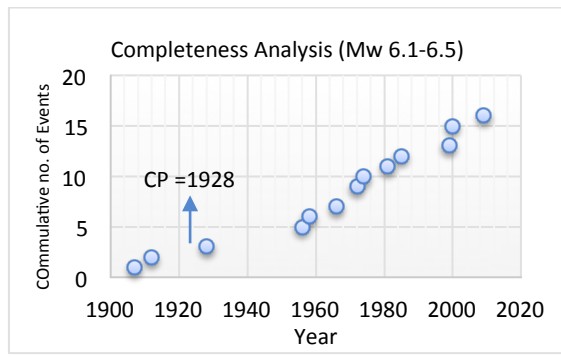

(Completeness period 87 years: 1928-2015)

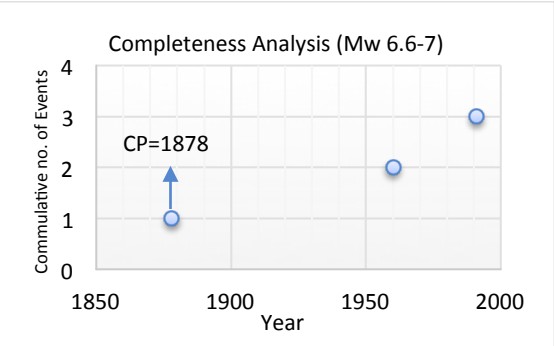

(Completeness period 137 years: 1878-2015)

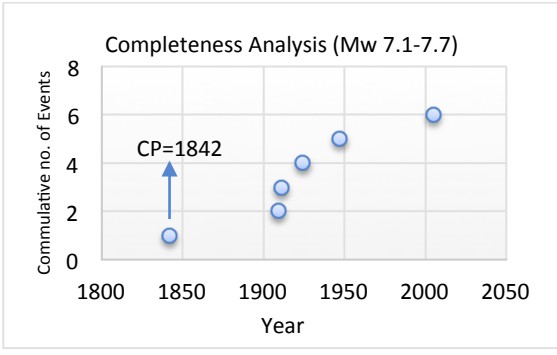

(Completeness period 173 years: 1842-2015)

**Figure 7**. Completeness Period for earthquake catalogue for specified band

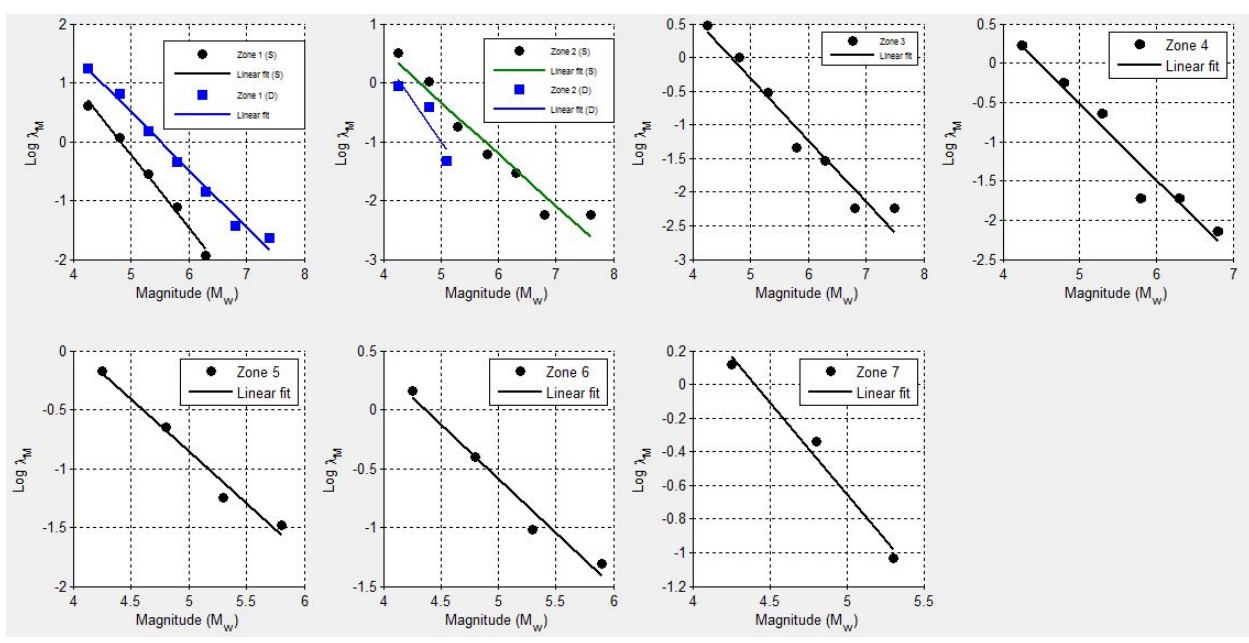

**Figure. 8** The graph $log_{\lambda_m} \sim M_w$ for seismic source parameters of seven zones

**4.4 Attenuation Relationships and Peak Ground Acceleration**

The attenuation relationships for a site are developed using substantial dataset information (Cotton et al.,

2006), however, these are not available for Pakistan because of the scarcity of available strong motion

data. The alternative to this is to use the already available attenuation relationships of other regions

having similar tectonic and geological conditions to Pakistan. In case of shallow earthquakes, the

candidate attenuation relationships for north Pakistan should be the one developed for the active tectonic

crustal earthquake region. Thus, the ground motion attenuation relationship of Akkar and Boomer (2010)

and Bore and Atkinson (2008) were used to calculate the PGA for shallow seismic sources. However, the

ground motion attenuation relationships of Lin and Lee (2008) and Kanno et al. (2006) developed for

subduction zones were used for deep seismic sources. The seismic hazard in term of PGA was then

calculated at bedrock site for different return periods, such as 50, 100, 250, 475 and 2500 years, as the

cumulative seismic hazard due to both shallow and deep seismic sources. The various GMPEs were

combined through logic tree approach and assigning equal weightages to each GMPE.  The ground

motions calculated were plotted in GIS environment to obtain the seismic hazard maps for these different

ground motion attenuation relationships.

1 **4.5 Seismic Hazard Maps**

2 The seismic hazard levels (**Table 5**), based on peak acceleration, defined in the BCP-SP (2007) were

3 considered as basis for zoning of the seismic hazard at bedrock:

**Table. 5** Seismic hazard levels used for seismic zoning, obtained from BCP-SP (2007)

| Seismic Hazard Level | Peak acceleration, (g) |
|---|---|
| Very low | < 0.08 g |
| Low | 0.08 - 0.16 g |
| Medium | 0.16 - 0.24 |
| High | 0.24 - 0.32 |
| Very high | > 0.32 g |

The seismic hazard maps for a return period of 475 years in case of shallow crustal earthquakes and deep

earthquakes for Peshawar District are reported in **Fig. 9** and **10** respectively. **Figure. 9** shows that for a

return period of 475 years, the predictive relationship of Akkar and Boomer, (2010) overestimate the

PGA value in comparison to that of Boore and Atkinson (2008), especially in the Northern parts of the

District. According to Arango et al. (2012), the distance scaling factor of the later appears to be more

adequate then the previous. Furthermore, **Table. 6** shows a slight comparison of both ground motion

prediction equations that suggests that in terms of $N_R$=Number of records, $T_{max}$= longest response period,

$M_w$ = moment magnitude and [R] = distance range, the prediction equation of Boore and Atkinson (2008)

is more appropriate and reliable than that of Akkar and Boomer (2010) for hazard assessment.

**Table. 4** Comparison of predictive equations used for shallow crustal earthquake (after, Arango, et al., 2012)

| Predictive equation | Tectonic Regime | Region | $N_R$ | $T_{max}$ | $M_w$ | [R] |
|---|---|---|---|---|---|---|
| Boore and Atkinson (2008) | Shallow crustal | Worldwide | 1574 | 10 | 5-8 | 0-200 |
| Akkar and Boomer (2010) | Shallow crustal | Europe/Middle east | 532 | 3 | 5-7.6 | 0-100 |

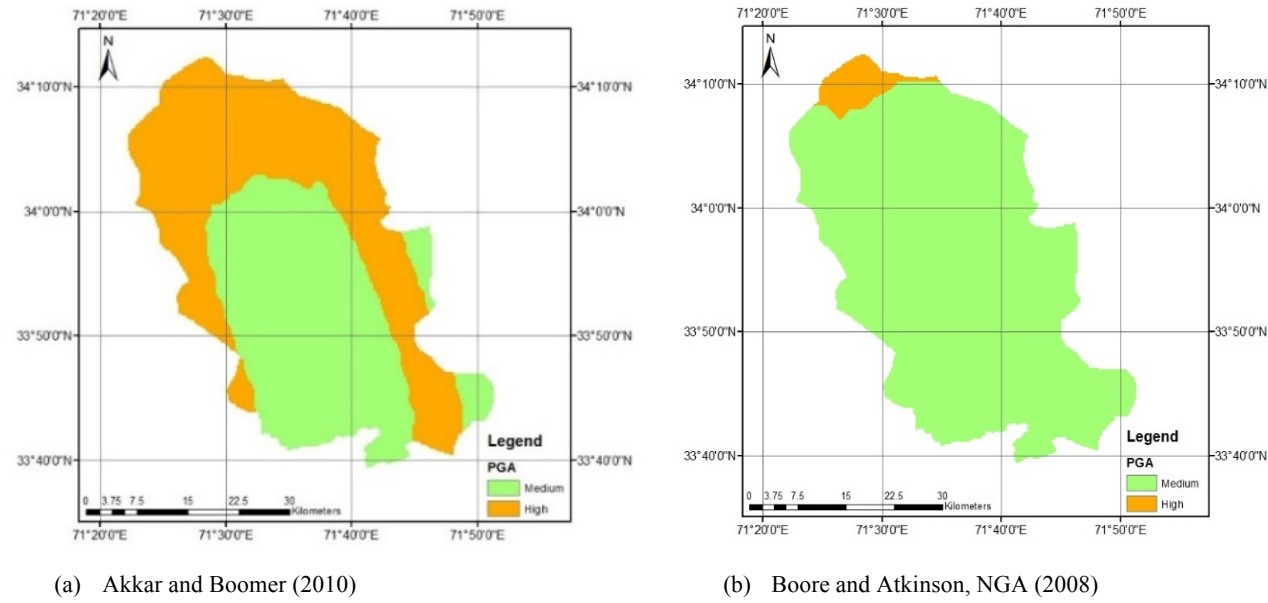

(a) Akkar and Boomer (2010)                    (b) Boore and Atkinson, NGA (2008)

**Figure. 9** Seismic hazard maps for shallow crustal earthquake using different attenuation equations

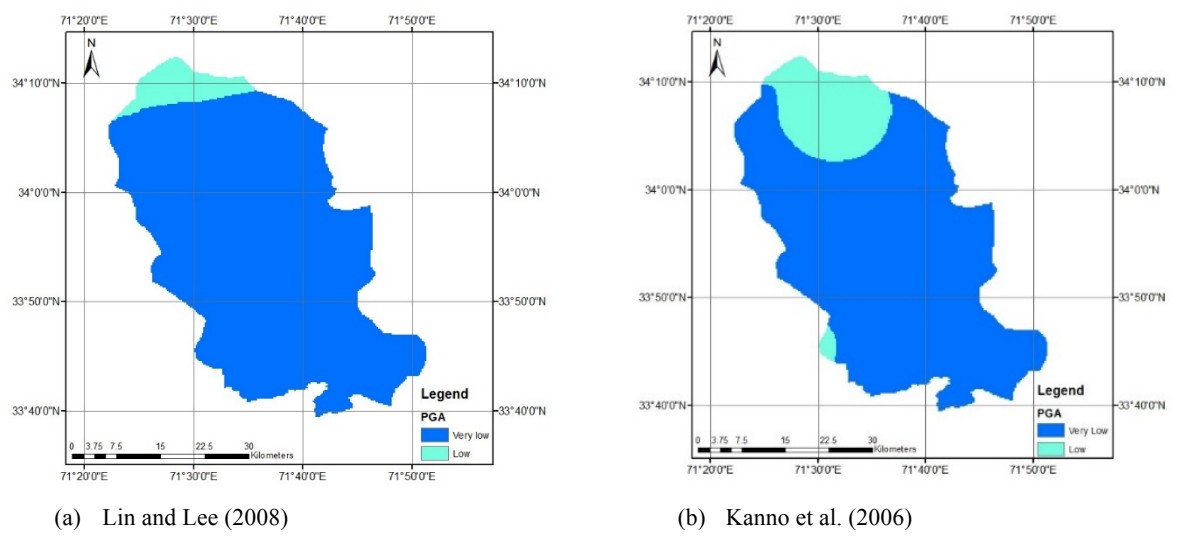

(a) Lin and Lee (2008)                    (b) Kanno et al. (2006)

**Figure. 10** Seismic hazard maps for deep subduction earthquake using different attenuation equations and for a return period of 475 years

**Figure. 10** shows, the seismic hazard maps for deep subduction earthquakes using the Lin and Lee (2008)
and Kanno et al. (2006) for a return period of 475 years. According to this **Fig. 10** both the attenuation
equations resulted in roughly similar seismic hazard for Peshawar District. Furthermore, it is also
evidenced from **Fig. 10** that, the inclusion of deep subduction zones in the seismic hazard does not
contribute significantly i.e., it remains low (0.08-0.16g) to very low (<0.08g). The cumulative seismic
hazard may slightly increase ground motion level, especially in the northern parts.

In probabilistic seismic hazard analysis (PSHA), one of the major sources of uncertainty is the epistemic uncertainty arising from the selection of predictive relationship. Thus, the different ground motion attenuation relationships already discussed were further used to find out the epistemic uncertainty in the seismic hazard analysis. This was accomplished through the logic tree approach, assigning equal weigthing factor to each GMPE (Fig. 11), the seismic hazard was combined from all the GMPEs.

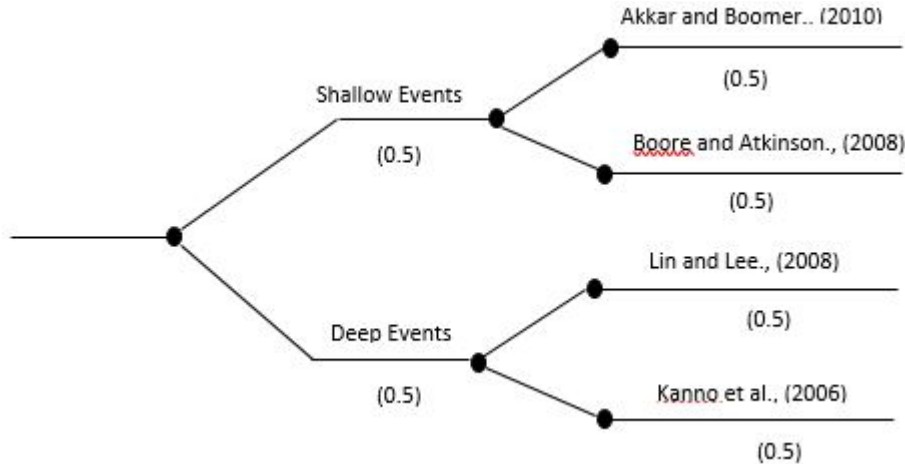

**Figure. 11** Logic Tree for incorporating epistemic uncertainty

**Figure. 12** shows the seismic hazard maps for shallow and deep events after incorporating the epistemic uncertainty. As can be seen in **Fig. 12a**, the seismic hazard of Peshawar District becomes balanced when the average of the seismic hazard calculated using the Akkar and Boomer (2010) and Boore and Atkinson (2008) were taken. The reason is that of providing equal weightage to both the predictive relationship in hazard analysis. The seismic hazard in case of deep subduction zone earthquake remains roughly the same after incorporating epistemic uncertainty (**Fig. 12b**). It can also be further concluded that the earthquake produced by deep subduction zone are not significant in term of seismic hazard and may be reasonably ignored. Thus, the shallow seismic sources are sufficient for seismic hazard assessment of Peshawar. The calculated seismic hazard map after incorporating epistemic uncertainty is compared with the hazard map from the BCP-2007. For the return period of 475 years, a close agreement between the two seismic hazard maps can be noticed (**Fig. 13**). After this check the seismic hazard maps for other return periods i.e. 50, 100, 250, 475 and 2500 years were prepared (**Fig. 14**), which may be used for seismic risk assessment. Hazard maps for various cases are reported in Appendix Fig. A1 through Fig. A8.

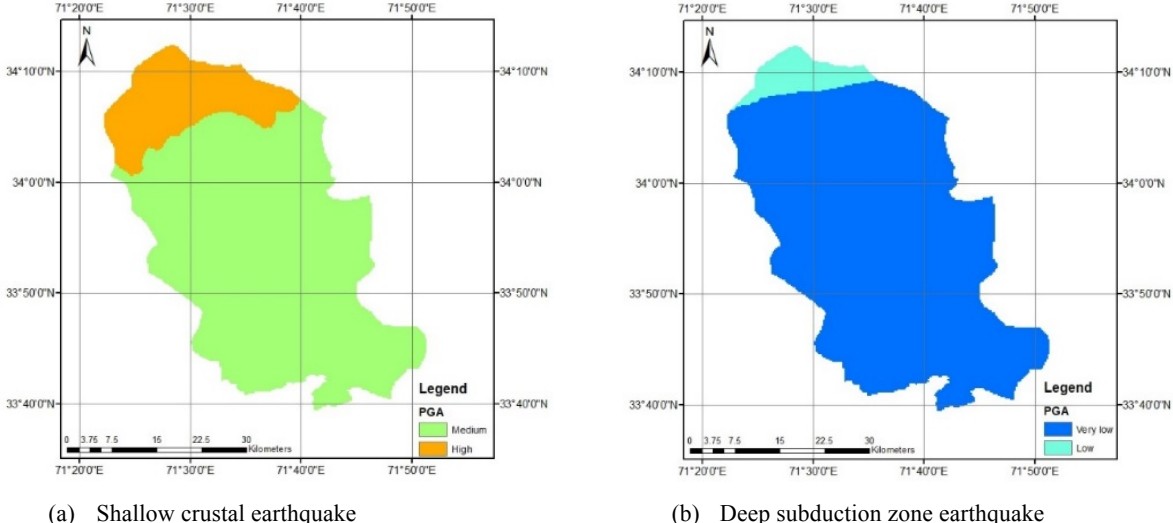

(a) Shallow crustal earthquake       (b) Deep subduction zone earthquake

**Figure. 12** Seismic hazard maps after incorporating epistemic uncertainty for 475 years return period

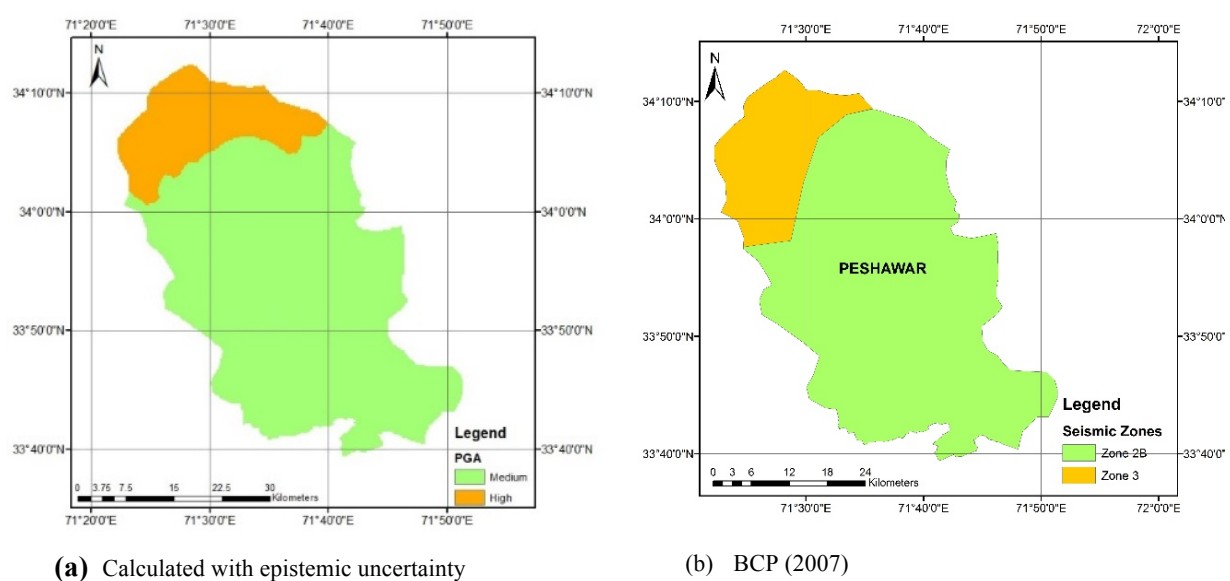

**(a)** Calculated with epistemic uncertainty       (b) BCP (2007)

**Figure. 13** Comparison of seismic hazard maps for a return period of 475 years

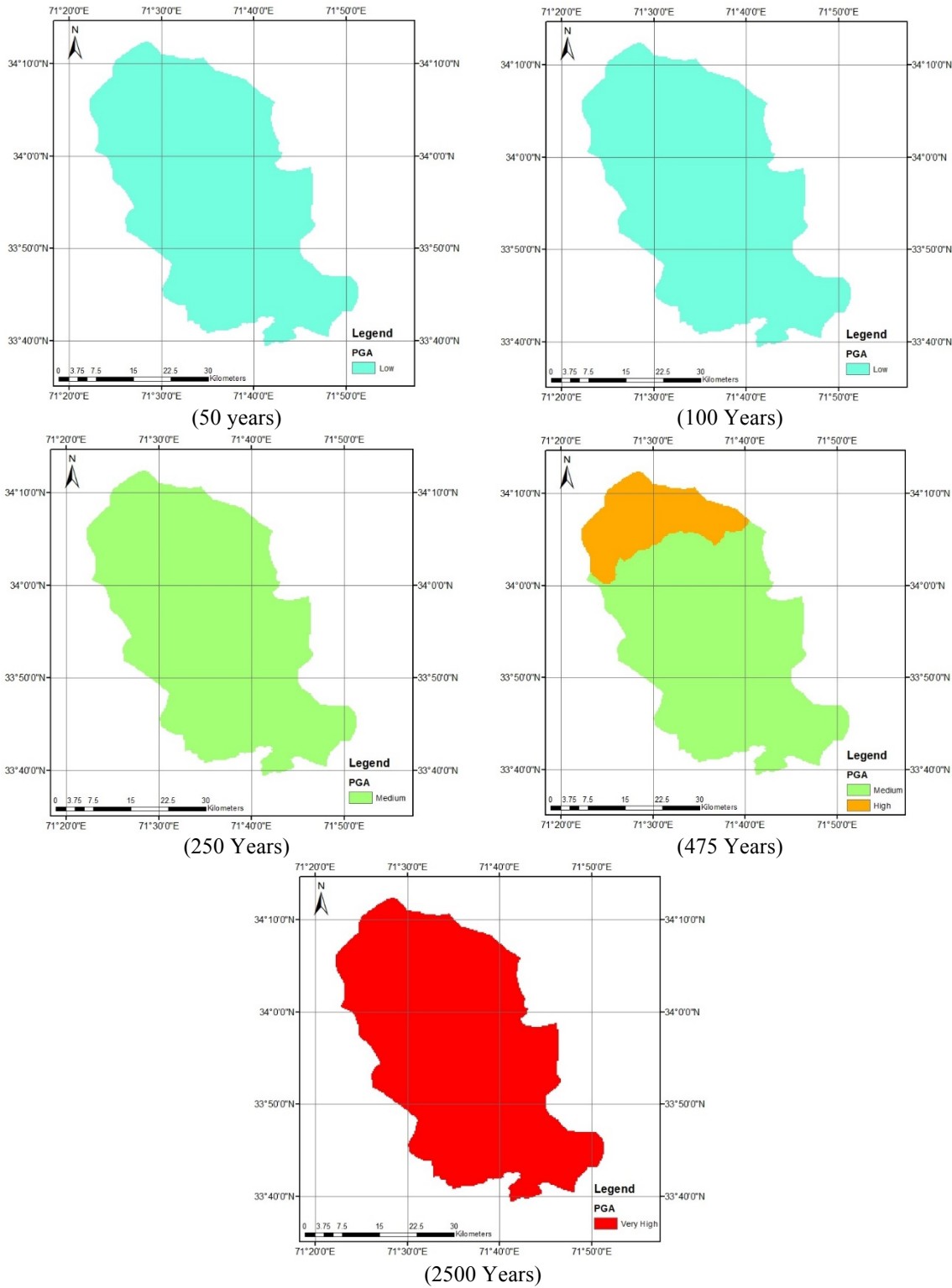

**Figure. 14 M**ean seismic hazard maps for various return periods i.e. 50, 250, 475, 2500 years, considering all GMPEs and both shallow and deep earthquake sources.

## 5. Conclusions and Recommendations

The following were concluded on the basis of literature review of past seismic hazard studies of Peshawar and classical PSHA conducted for Peshawar in the present study:

- The selection of appropriate ground motion prediction equation is crucial in defining the seismic hazard of Peshawar District. In case of shallow crustal earthquake, the predictive relationship of Akkar and Boomer (2010) provide higher estimate of the PGA value in comparison to that of Boore and Atkinson (2008). The distance-scaling factor of the later appears to be the reason for this disparity between the two models.

- The inclusion of deep subduction earthquakes does not add significantly to hazard and may be neglected in term of seismic hazard. Therefore, these are only the shallow crustal earthquakes that contribute to the seismic hazard of Peshawar District. However, recent earthquakes in Peshawar from deep sources earthquakes has caused widespread destruction in various parts of the district. This raises concern for the existing GMPEs and the classical PSHA procedure to simulate such effects.

- The epistemic uncertainty was used by providing equal weightage to the attenuation equation of Akkar and Boomer (2010) and Boore and Atkinson (2008). The mean seismic hazard map thus produced was balanced and was found in a close agreement with the design base seismic hazard given in the BCP-SP (2007) for bedrock hazard. However, the BCP places Peshawar in Zone 2B, which is reasonable for most of the locations but it underestimates ground motions especially in northern parts of the District.

- The mean seismic hazard calculated for Peshawar was also compared with the previous studies (Table 6). It can be observed that the seismic hazard performed by independent researchers suggests an average PGA equal to about 0.24g, which is in agreement with the PGA = 0.24g given in the BCP-SP (2007) for seismic Zone 2b (0.16g to 0.24g) for bedrock. The present PSHA study performed using the most up-to-date earthquake catalogue, recent GMPEs and considering both the shallow and deep seismic sources confirmed the validity of seismic hazard given in the BCP-SP (2007). It is worth mentioning that the calculated mean hazard may be approximated as the $50^{th}$ percentile seismic hazard. **Table 6** reports that recent hazard studies considering the fault sources have resulted in larger estimate of seismic hazard that places Peshawar in seismic Zone 3 and Zone 4, however, the idealization of seismic sources as discrete faults for Peshawar are not reliable due to the scarcity of detailed information regarding the fault sources in Northern Pakistan. This higher seismic hazard is not justified by the earthquake history of Peshawar.

**Table 6.** Placement of Peshawar based on the present study: classical PSHA with areal sources, considering both shallow and deep earthquakes. Listed in ascending order.

| S. No. | Authors | PGA (g) |
|:---:|:---|:---:|
| 1 | Shah et al. (2019) | 0.06 |
| 2 | Bhatia et al. (1999) | 0.10 – 0.15 |
| 3 | Mona Lisa et al. (2007) | 0.15 |
| 4 | Zhang et al. (1999) | 0.16 – 0.24 |
| 5 | Rafi et al. (2012) | 0.17 |
| 6 | Ahmad et al. (2019) | 0.16 to 0.24 |
| **7** | **Present Study** | **0.16 – 0.32** |
| 8 | Hashash et al. (2012) | 0.20 – 0.40 |
| 9 | Şeşetyan et al. (2018) | 0.30 – 0.40 |
| 10 | Khaliq et al. (2019) | 0.32 – 0.34 |
| 11 | Zaman and Warnitchai (2012) | 0.33 – 0.40 |
| 12 | Waseem et al. (2020) | 0.33 |
| 13 | Waseem et al. (2018) | 0.38 |

It is worth mentioning that the focus of the present study was to provide the base maps for seismic hazard in Peshawar. Site-specific soil properties were not known, therefore, it was not addressed in the present study. Alternatively, the code suggests amplification factors for various soils from Type C to Type E as per NEHRP soil classification. This may be considered to amplify/de-amplify the seismic hazard provided in the present study.

**Acknowledgement**

This paper has been produced from MSc research work of Seismic Microzonation of Peshawar, Pakistan. The authors are grateful to the reviewer for their constructive remarks that improved quality of the manuscript.

**Data Availability Statement**

All data, models, or code generated or used during the study are available from the corresponding author by request (Naveed Ahmad, naveed.ahmad@uetpeshawar.edu.pk). Items, which may be requested: earthquake catalogue (raw and processed data), excel sheets used for G-R parameters derivation, CRISIS input files, etc.

**Author Contribution**

Naveed Ahmad has contributed as the advisor/supervisor of the research, in the analysis of earthquake
catalogue for derivation of seismic parameters, selection of GMPEs, preparation of input files for CRISIS
program and paper drafting. Usman Khan has contributed in the compilation of earthquake catalogue and
processing of data, analysis of hazard through CRISIS program and developing hazard maps. Khalid
Mahmood has contributed in the data organization, literature review and integration of work tasks and
preparation of initial paper draft. Qaiser Iqbal has contributed in selecting/designing seismic sources for
both shallow and deep earthquakes, assignment of source parameters in CRISIS program, output data
compilation and result plotting.

**Competing Interest**

The authors do not report any potential conflict of interest.

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

1      **APPENDIX:**

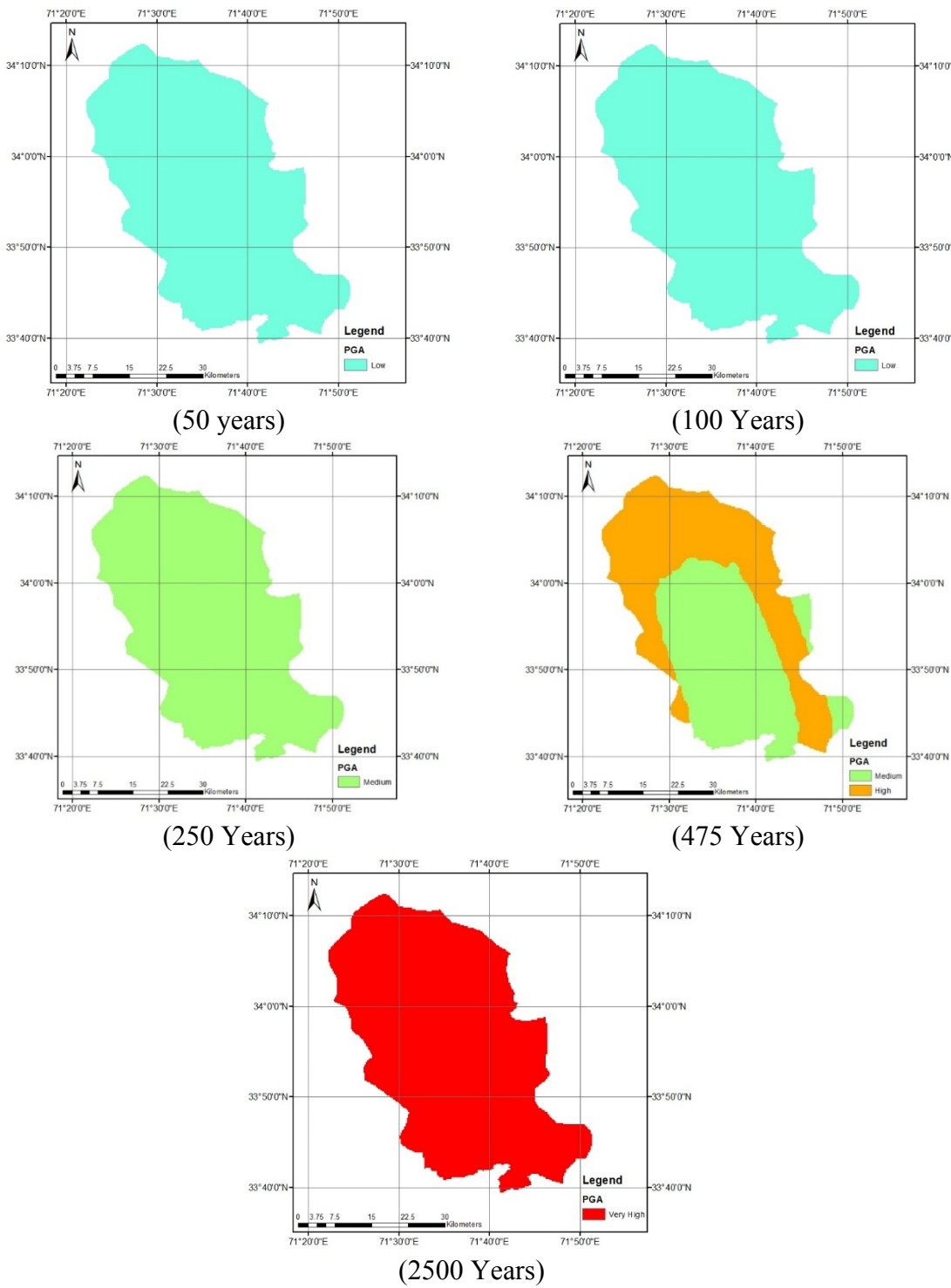

(50 years)                              (100 Years)

(250 Years)                             (475 Years)

(2500 Years)

3          Fig A1. Calculated seismic hazard maps using Akkar and Boomer, 2010 GMPE for shallow earthquakes

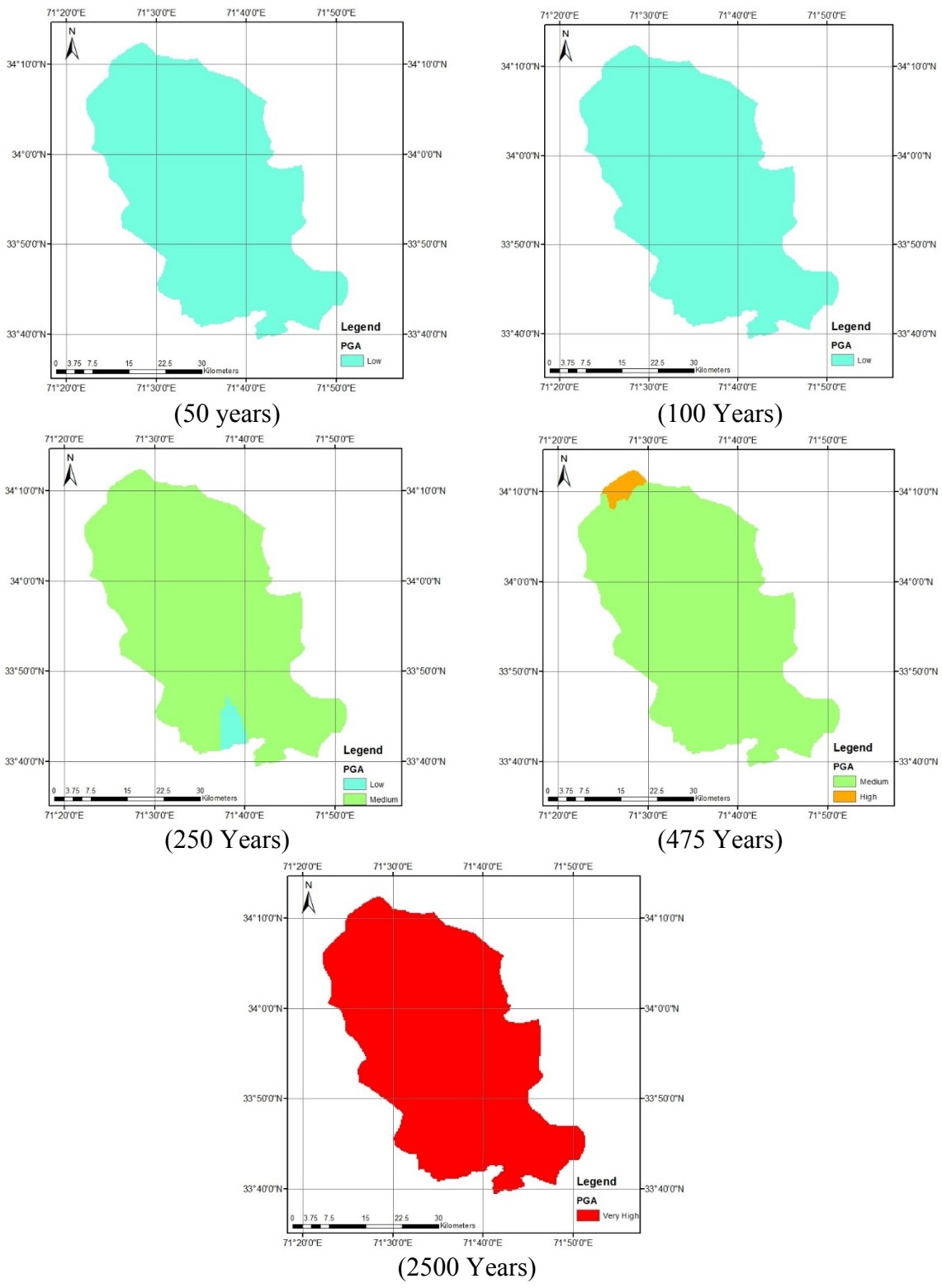

(50 years)
(100 Years)
(250 Years)
(475 Years)
(2500 Years)

Fig A2. Calculated seismic hazard map using Bore and Atkinson, NGA 2008 GMPE for shallow earthquakes

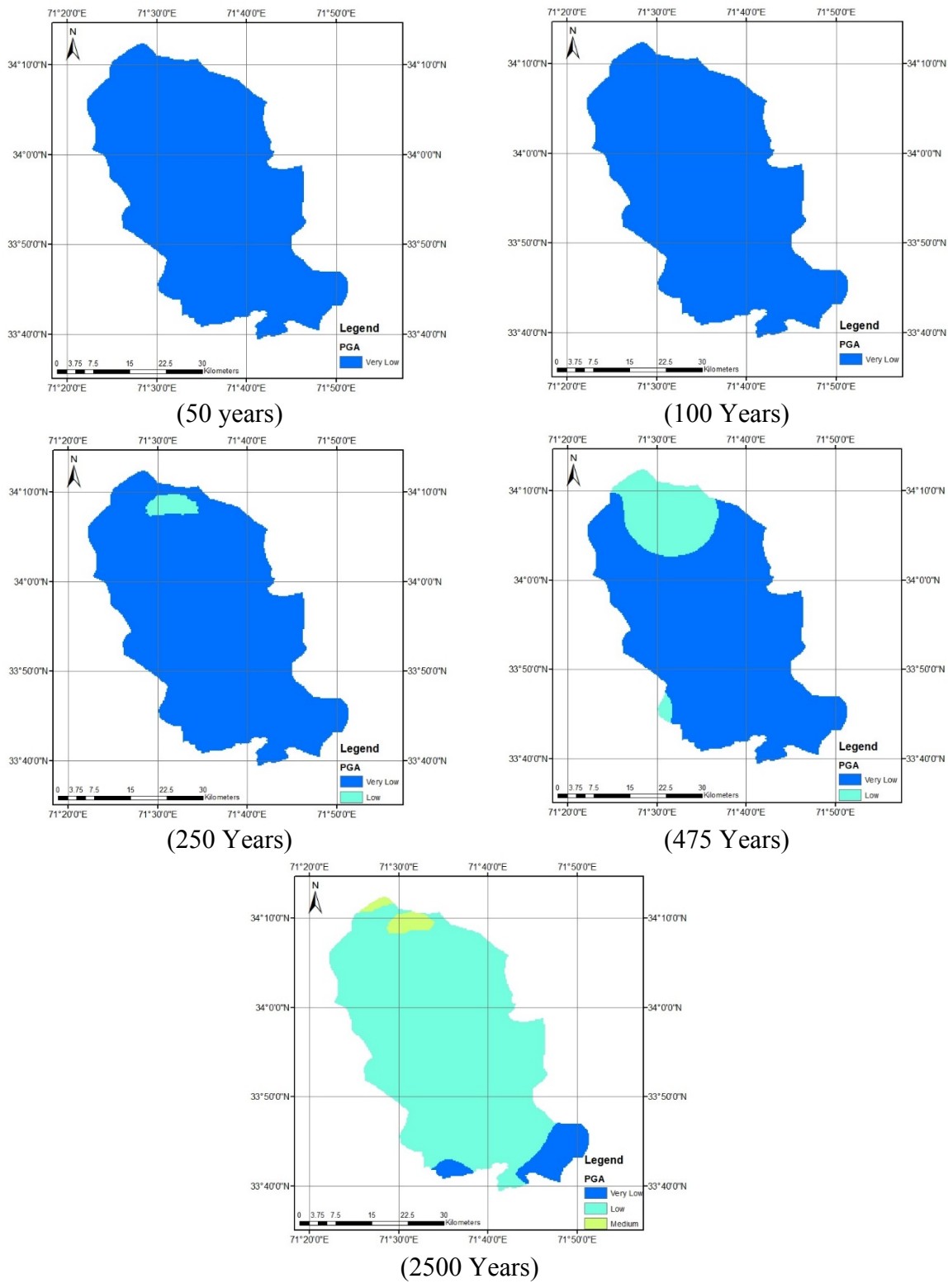

Fig A3. Calculated seismic hazard map using Kanno et al, 2006 GMPE for Deep earthquakes

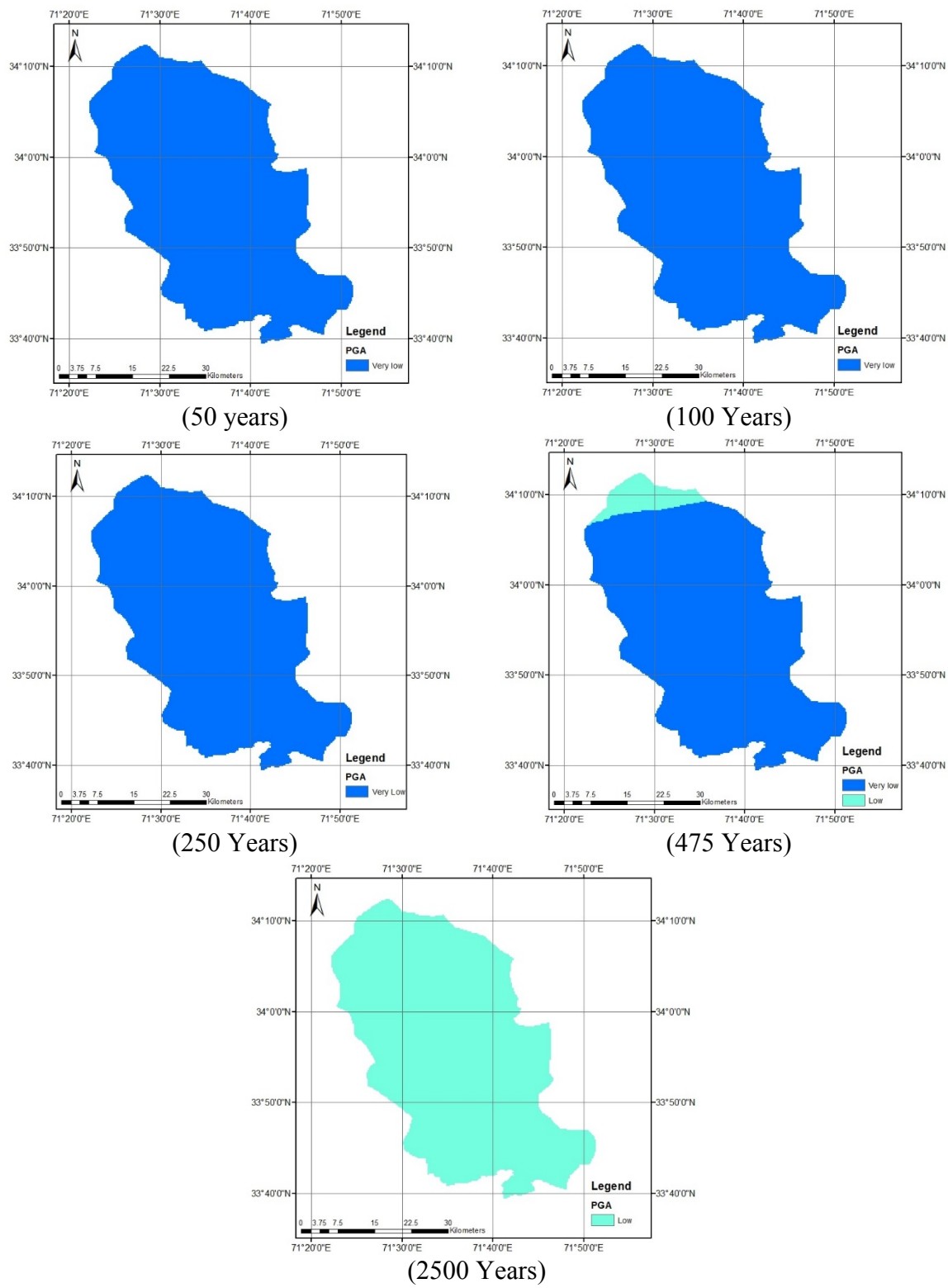

(50 years)  (100 Years)

(250 Years)  (475 Years)

(2500 Years)

2          Fig A4 Calculated seismic hazard map Lin and Lee, 2008 GMPE for Deep earthquakes

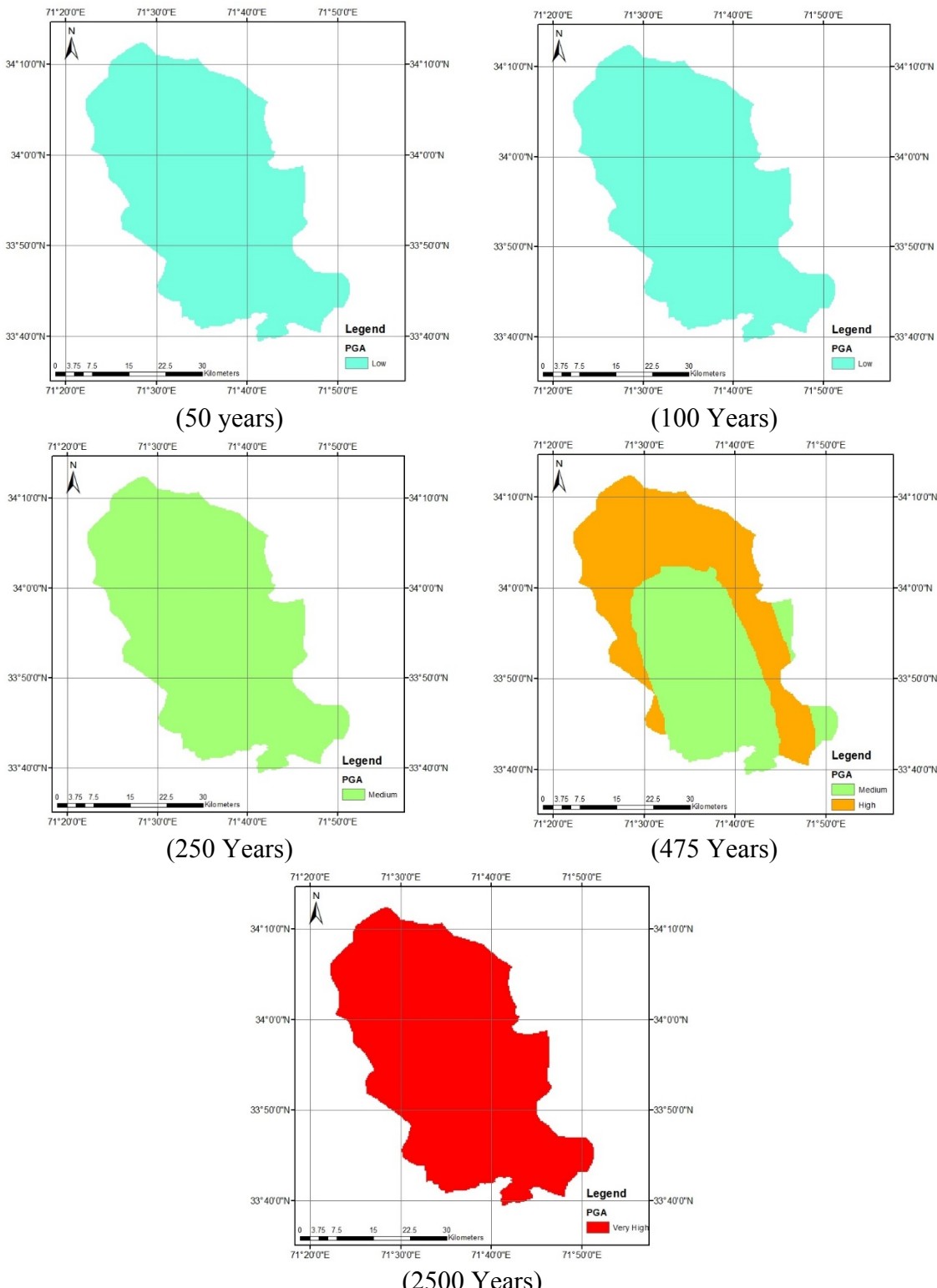

(50 years)

(100 Years)

(250 Years)

(475 Years)

(2500 Years)

Fig A5. Calculated seismic hazard map considering Akkar and Boomer, 2010 for shallow earthquakes and Kanno et al, 2006 for Deep earthquakes

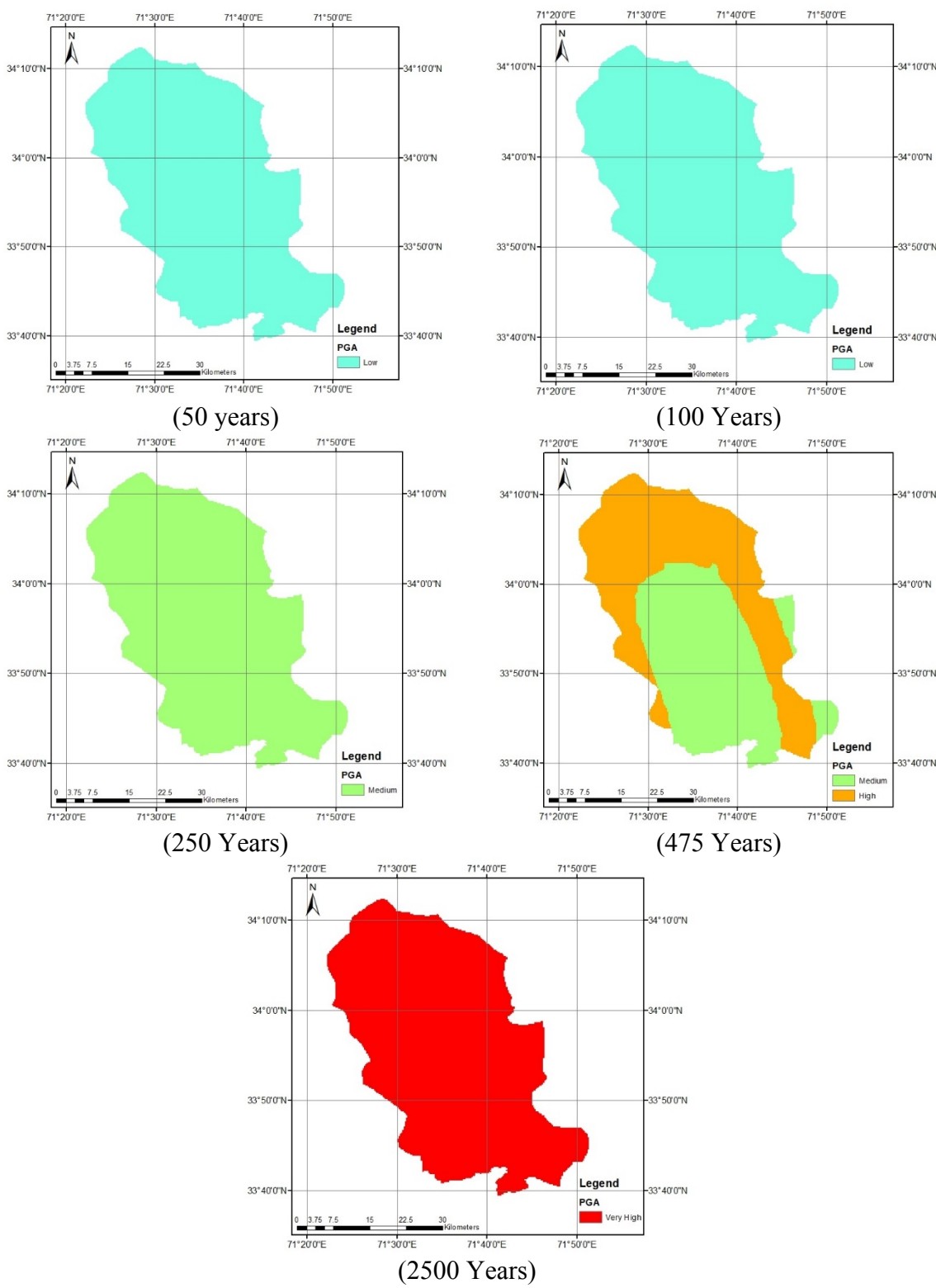

Fig A6. Calculated seismic hazard map considering Akkar and Boomer, 2010 for shallow earthquakes and Lin and Lee, 2008 for Deep earthquakes

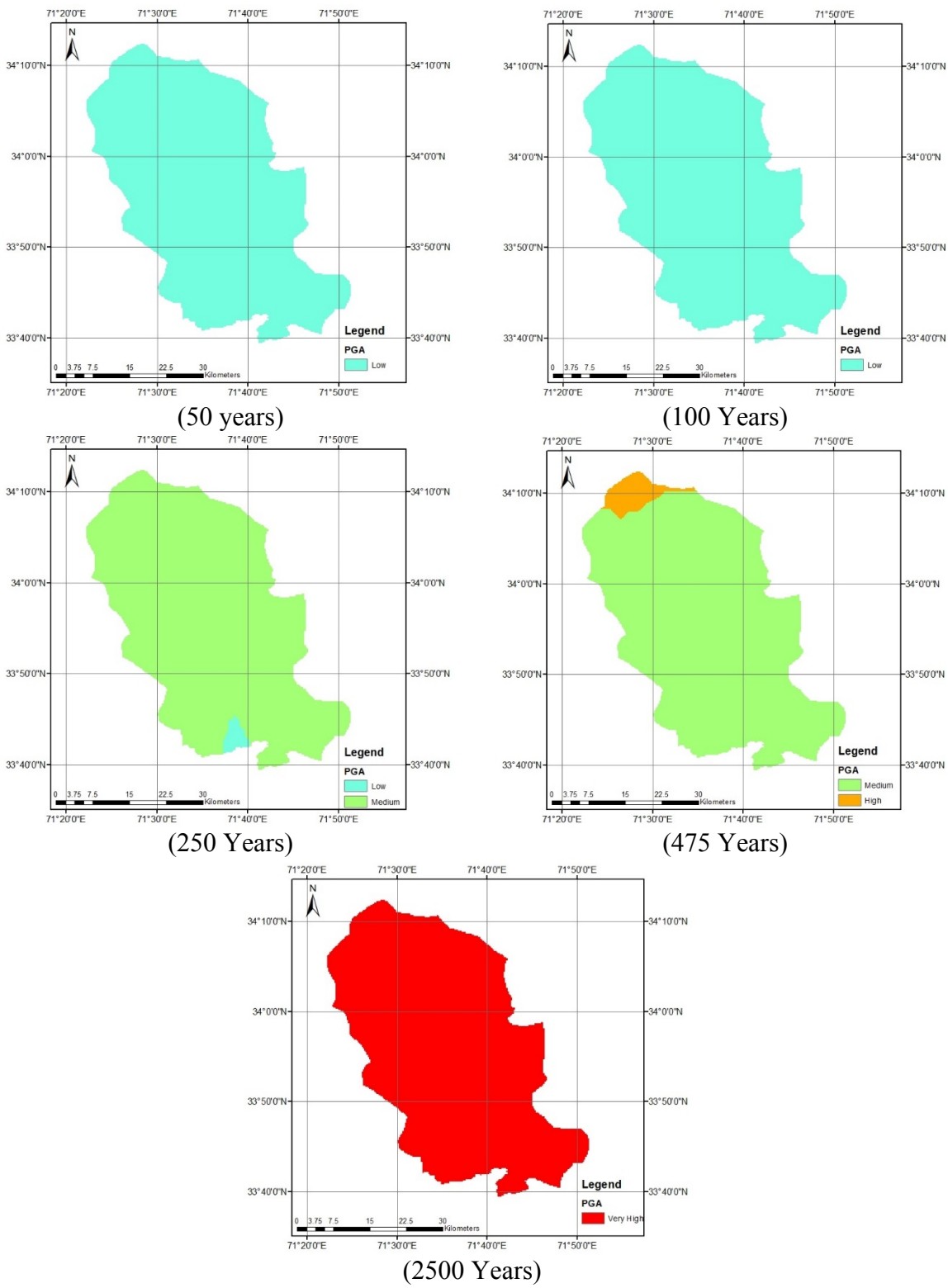

Fig A7. Calculated seismic hazard map considering Bore and Atkinson, NGA 2008 for shallow earthquakes and Kanno et al, 2006 for Deep earthquakes

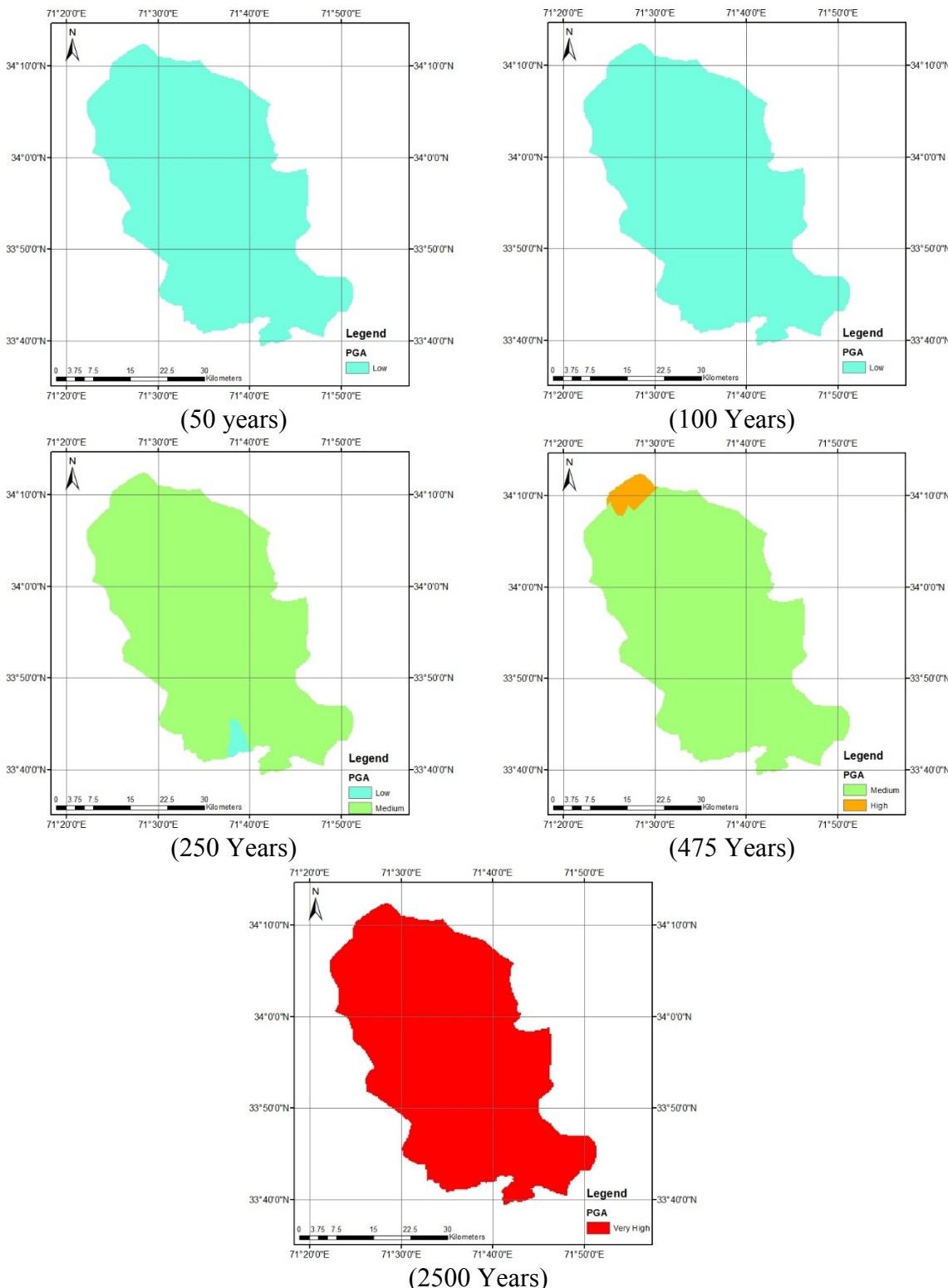

Fig A8. Calculated seismic hazard map considering Bore and Atkinson, NGA 2008 for shallow earthquakes and Lin and Lee, 2008 for Deep earthquakes