# Peer review of "SEISMIC HAZARD MAPS OF PESHAWAR DISTRICT FOR VARIOUS RETURN PERIODS"

_Natural Hazards and Earth System Sciences, 2019_

## Referee Comment (RC1) · Anonymous Referee #1 · 22 Dec 2019

The article by Khalid et al. (2019) presents the probabilistic seismic hazard analysis of Peshawar district of Pakistan. The authors have used the Cornell- McGuire approach to carry out the assessment and four different ground motion prediction equations have been used combined with the logic tree and the final maps are presented for peak ground acceleration (PGA) values. I have gone through the article and noted the following observations: 1. There is very little information is provided about the compilation of earthquake catalog, the historical era events may be discussed in the article and a small discussion on its compilation is required to be included. 2. Completeness analysis and declustering of catalog is missing, both catalog completeness and declustering should be discussed, as these seriously effect the hazard values. 3. Several important studies concerning Peshawar, Khaliq et al. (2018); Waseem et al. (2018); Sesetayn et

al. (2018) Ahmad et al. (2019) have been missed by the authors, they should be cited in the article, where required. 4. The results of this study should be discussed and compared with the similar studies

---

## Referee Comment (RC2) · Anonymous Referee #2 · 23 Dec 2019

The authors present an interesting work on probabilistic seismic hazard assessment of the Peshawar district in Pakistan using seven areal source zones. The paper is well written and easy to follow. However, there are some issues that need to addressed before the paper can be accepted for publication as follows: 1. Please justify why you opted for areal sources surpassing point and line sources. Or, why you did not opt for a combination of all? 2. The attenuation relationship you are using is not from the Himalaya. Would it be possible to calibrate some subduction GMPE with the available records from Pakistan to obtain more realistic results? 3. As you contrasted yourself, the effects of deep earthquakes were pronounced recently in Pakistan yet you did not include the effect. Could you please reframe the logic tree in any way to incorporate this? 4. The source zones are somehow interesting too. For instance, why zones 5,

6, and 7 have quite limited data? Could you please elucidate your zoning scheme? 5. As you have prepared the hazard maps for bedrock, I request you to consider hazard maps on the surface too [if possible]. If you have some site response/amplification studies, it would be interesting and also useful for the structural earthquake engineering communities. Please comment. 6. Please fix some grammatical bugs present in the manuscript.

---

## Author Comment (AC1) · 4 Mar 2020

The authors are very grateful to the Editors and Associate Editors for the kind consideration and possible publication of our article in the Natural Hazards and Earth System Sciences. The authors would like to thank all reviewers for suggesting improvements for the manuscript. Point-wise reply/answer to each comment is provided below (comments are shown in BOLD, answers are shown in REGULAR). All suggestions have been addressed, but still if reviewers have any other point/reservation, the authors are happy to incorporate. Furthermore, the authors appreciate the editors and reviewers for the timely handling of review process. The responses are also attached in .pdf format.

[Figure]

REVIEWER 1

1. There is very little information is provided about the compilation of earthquake catalog, the historical era events may be discussed in the article and a small discussion on its compilation is required to be included.

Response: The reviewer is thanked for this suggestion, the revised manuscript now incorporated the requested information. The catalogue was compiled till 2015 that also included historical earthquake data from Ambrasey (2000) and Ambrasey and Douglas (2004). Below is the priority list for the catalogue homogenization and correctness to remove the duplicate events:

Priority Data Source 1 Ambrasey & Douglas (2004) 2 Ambrasey (2000) 3 ISC 4 GCMT 5 NGDC 6 USGS

2. Completeness analysis and declustering of catalog is missing, both catalog completeness and declustering should be discussed, as these seriously effect the hazard values.

Response: The authors fully agree with the reviewer. The requested information is provided in the revised manuscript.

3. Several important studies concerning Peshawar, Khaliq et al. (2018); Waseem et al. (2018); Sesetayn et al. (2018) Ahmad et al. (2019) have been missed by the authors, they should be cited in the article, where required.

Response: The authors thank the reviewer for this suggestion, the revised manuscript now incorporated the mentioned references.

4. The results of this study should be discussed and compared with the similar studies.

Response: The authors fully agree to the reviewer, the revised manuscript now compared the outcomes with relevant studies.

Please also note the supplement to this comment:
https://www.nat-hazards-earth-syst-sci-discuss.net/nhess-2019-299/nhess-2019-299-AC1-supplement.pdf

―――――――――――――――――――――

[Figure]

**Supplement:**

**RESPONSE TO THE EDITOR'S COMMENTS**

The author is very grateful to the Editors and Associate Editors for the kind consideration and possible publication of our article in the European Journal of Environmental and Civil Engineering. The authors would like to thank all the reviewer(s) for suggesting improvements for the manuscript. Point-wise reply/answer to each comment is provided below (comments are shown in BOLD, answers are shown in REGULAR and modifications/added lines are shown in RED COLOR). All suggestions have been addressed, but still if reviewer(s) have any other point/reservation, the authors are happy to incorporate. Furthermore, the authors are very thankful and appreciate the associate editors and reviewers for the timely handling of review process. The revised paper is provided in both formats (with Track Changes and Final version).

**REVIEWER 1**

| Sr. No. | Questions with answers | Clarification made or Changes Incorporated |
|---------|------------------------|--------------------------------------------|
| 1 | **1. The abstract should be totally written, such that to be shorter and present a summary of all the parts of the paper including the results and achievements.**

 *The reviewer is thanked for the suggested improvement, which has been made in the revised paper.* | YES |
| 2 | **2. The English need to be improved. For instance, see the last sentence in Page 1.**

 *The reviewer is thanked for the indicated corrections, which have been made in the revised version of the paper. Furthermore, the manuscript is re-visited for English writing improvement.* | YES |

| Sr. No. | Questions with answers | Clarification made or Changes Incorporated |
|---|---|---|
| 3 | **3. The sentences are lengthy in many parts of the paper.. Understanding of such sentences is hard.**

*The reviewer is thanked for the suggested improvement, which has been made in the revised version of the paper.* | YES |
| 4 | **4. In Page 3, a two story frame is considered representative of hospitals, shopping malls, and schools. With attention to different geometry and loadings of these three types of buildings, more explanation regarding the representative frame is essential.**

*The reviewer is thanked for the suggestion improvement, which has been made in the revised paper. For the kind information, the RC frames studied in the present research are representative of commercial and public buildings. Because construction of hospital, plazas, schools etc., is carried out using RC frames.* | YES |
| 5 | **5. The caption of Fig. 3 should include the explanations under the several figures in Fig. 3. (Similar problem also exist in Figs . 5 and 6)**

*The reviewer is thanked for the suggested improvement, which has been made in the revised paper.* | YES |
| 6 | **6. An additional lines seems existing at the end of the caption of Fig. 4.**

*The reviewer is thanked for pointing to this, the second line describe the fitting shown in the plot.* | Comments are provided herein to clarify the author's intention. |
| 7 | **7. The numbers in Figs. 5-7 are too small to be read.**

*The reviewer is thanked for the suggested improvement, which has been made in the revised paper.* | YES |
| 8 | **8. Some legend should express the meaning of blue and red colors in Figs. 5-7**

*The reviewer is thanked for the suggested improvement, which has been made in the revised paper.* | YES |

| Sr. No. | Questions with answers | Clarification made or Changes Incorporated |
|---|---|---|
| 9 | **9. It is not clear which seismic code is under consideration, neither from the text nor from the references.**

*The reviewer is thanked for the suggested improvement, which has been made in the revised paper.* | YES |
| 10 | **10. I could not find any thing new in the paper even after reviewing the "Conclusions" section.**

*The reviewer is thanked for pointing to this. The authors have investigated deficient RC frames having weaker beam-column joints. The authors have derived force reduction factor and displacement amplification factor for RC frames with weaker joints. These parameters are not available for the considered structures in the available literature. Further, the authors have developed and applied a simplified static force-base procedure for seismic analysis and vulnerability assessment of similar like structures. The proposed procedure and the derived seismic response parameters will enable engineers for the preliminary vulnerability assessment of RC frame structures having weaker beam-column joints. This clarified also in the revised manuscript.* | Comments are provided herein to clarify the author's intention. |

---

## Author Comment (AC2) · 4 Mar 2020

The authors are very grateful to the Editors and Associate Editors for the kind consideration and possible publication of our article in the Natural Hazards and Earth System Sciences. The authors would like to thank all reviewers for suggesting improvements for the manuscript. Point-wise reply/answer to each comment is provided below (comments are shown in BOLD, answers are shown in REGULAR). All suggestions have been addressed, but still if reviewers have any other point/reservation, the authors are happy to incorporate. Furthermore, the authors appreciate the editors and reviewers for the timely handling of review process. Point-wise response is also attached in .pdf format.

[Figure]

REVIEWER 2

1. Please justify why you opted for areal sources surpassing point and line sources. Or, why you did not opt for a combination of all?

The reviewer is thanked for pointing to this. The available catalogue and seismicity pattern is quite scattered and not very specific to suggest taking line/fault sources. There are not enough data (fault source information) to support line sources based hazard assessment. Moreover, due to the scattered seismicity pattern and large number of small fault-lines, the definition of fault line becomes challenging. Areal sources are reasonable approximation to idealize seismic sources and used in many similar studies. Few studies have used fault sources but that have resulted in very high seismic hazard, which is not justified by the history of earthquakes in Peshawar.

2. The attenuation relationship you are using is not from the Himalaya. Would it be possible to calibrate some subduction GMPE with the available records from Pakistan to obtain more realistic results?

The authors fully agree with the reviewer. Pakistan doesn't have specific GMPEs of its own, therefore, GMPEs from other similar region are adopted. The selected GMPEs were earlier tested in prediction of ground motion for selected earthquake events. The GMPEs, which have shown relatively better performance, were selected for hazard analysis. This is clarified in the revised manuscript.

3. As you contrasted yourself, the effects of deep earthquakes were pronounced recently in Pakistan yet you did not include the effect. Could you please reframe the logic tree in any way to incorporate this?

The authors fully agree with this. Deep sources were included in the hazard assessment, however, this had little influence on the final hazard maps. Possible reason seems to be the limited earthquake catalogue and the foreign GMPEs, which were not specific to the region. Also, the GMPEs lack to take into account the site effects common in Peshawar valley due to deep earthquakes. We didn't find any other alternative to manifest the deep sources effects accurately, however, we put this as a question for others to address.

4. The source zones are somehow interesting too. For instance, why zones 5, 6, and 7 have quite limited data? Could you please elucidate your zoning scheme?

The authors fully agree to the reviewer, however, the seismic sources used for shallow earthquakes were those obtained from the Building Code of Pakistan – Seismic Provisions. The deep sources were selected in consultation with the National Center of Excellence in Geology, Peshawar. Since, deep sources are not studied before for Peshawar. This is clarified in the revised manuscript.

5. As you have prepared the hazard maps for bedrock, I request you to consider hazard maps on the surface too [if possible]. If you have some site response/amplification studies, it would be interesting and also useful for the structural earthquake engineering communities. Please comment.

The authors fully agree to the reviewer, however, the focus of present study was to provide the base maps for hazard. Site-specific soil was not known so it was not addressed in the hazard assessment. Alternatively, the code suggests amplification factors for various soil from Type C to Type E as per NEHRP soil classification.

6. Please fix some grammatical bugs present in the manuscript.

The authors thank the reviewer for this suggestion, the revised manuscript is re-visited for English writing improvement.

Please also note the supplement to this comment:
https://www.nat-hazards-earth-syst-sci-discuss.net/nhess-2019-299/nhess-2019-299-AC2-supplement.pdf
* * *
2019-299, 2019.

**Supplement:**

**RESPONSE TO THE EDITOR'S COMMENTS**

The author is very grateful to the Editors and Associate Editors for the kind consideration and possible publication of our article in the European Journal of Environmental and Civil Engineering. The authors would like to thank all the reviewer(s) for suggesting improvements for the manuscript. Point-wise reply/answer to each comment is provided below (comments are shown in BOLD, answers are shown in REGULAR and modifications/added lines are shown in RED COLOR). All suggestions have been addressed, but still if reviewer(s) have any other point/reservation, the authors are happy to incorporate. Furthermore, the authors are very thankful and appreciate the associate editors and reviewers for the timely handling of review process. The revised paper is provided in both formats (with Track Changes and Final version).

**REVIEWER 1**

| Sr. No. | Questions with answers | Clarification made or Changes Incorporated |
|---------|------------------------|--------------------------------------------|
| 1 | **1. The abstract should be totally written, such that to be shorter and present a summary of all the parts of the paper including the results and achievements.**

 *The reviewer is thanked for the suggested improvement, which has been made in the revised paper.* | YES |
| 2 | **2. The English need to be improved. For instance, see the last sentence in Page 1.**

 *The reviewer is thanked for the indicated corrections, which have been made in the revised version of the paper. Furthermore, the manuscript is re-visited for English writing improvement.* | YES |

| Sr. No. | Questions with answers | Clarification made or Changes Incorporated |
|---|---|---|
| 3 | **3. The sentences are lengthy in many parts of the paper.. Understanding of such sentences is hard.**

*The reviewer is thanked for the suggested improvement, which has been made in the revised version of the paper.* | YES |
| 4 | **4. In Page 3, a two story frame is considered representative of hospitals, shopping malls, and schools. With attention to different geometry and loadings of these three types of buildings, more explanation regarding the representative frame is essential.**

*The reviewer is thanked for the suggestion improvement, which has been made in the revised paper. For the kind information, the RC frames studied in the present research are representative of commercial and public buildings. Because construction of hospital, plazas, schools etc., is carried out using RC frames.* | YES |
| 5 | **5. The caption of Fig. 3 should include the explanations under the several figures in Fig. 3. (Similar problem also exist in Figs . 5 and 6)**

*The reviewer is thanked for the suggested improvement, which has been made in the revised paper.* | YES |
| 6 | **6. An additional lines seems existing at the end of the caption of Fig. 4.**

*The reviewer is thanked for pointing to this, the second line describe the fitting shown in the plot.* | Comments are provided herein to clarify the author's intention. |
| 7 | **7. The numbers in Figs. 5-7 are too small to be read.**

*The reviewer is thanked for the suggested improvement, which has been made in the revised paper.* | YES |
| 8 | **8. Some legend should express the meaning of blue and red colors in Figs. 5-7**

*The reviewer is thanked for the suggested improvement, which has been made in the revised paper.* | YES |

| Sr. No. | Questions with answers | Clarification made or Changes Incorporated |
|---|---|---|
| 9 | **9. It is not clear which seismic code is under consideration, neither from the text nor from the references.**

*The reviewer is thanked for the suggested improvement, which has been made in the revised paper.* | YES |
| 10 | **10. I could not find any thing new in the paper even after reviewing the "Conclusions" section.**

*The reviewer is thanked for pointing to this. The authors have investigated deficient RC frames having weaker beam-column joints. The authors have derived force reduction factor and displacement amplification factor for RC frames with weaker joints. These parameters are not available for the considered structures in the available literature. Further, the authors have developed and applied a simplified static force-base procedure for seismic analysis and vulnerability assessment of similar like structures. The proposed procedure and the derived seismic response parameters will enable engineers for the preliminary vulnerability assessment of RC frame structures having weaker beam-column joints. This clarified also in the revised manuscript.* | Comments are provided herein to clarify the author's intention. |

---

## Author Response (AR1)

**RESPONSE TO THE EDITOR'S COMMENTS**

The authors are very grateful to the Editors and Associate Editors for the kind consideration and possible publication of our article in the Natural Hazards and Earth System Sciences. The authors would like to thank all reviewers for suggesting improvements for the manuscript. Point-wise reply/answer to each comment is provided below (comments are shown in BOLD, answers are shown in REGULAR). All suggestions have been addressed, but still if reviewers have any other point/reservation, the authors are happy to incorporate. Furthermore, the authors appreciate the editors and reviewers for the timely handling of review process.

| Sr.
No. | C                                                                                                                                                                                                                                          | Questions with answers                                                                                                                                                                                                                                                                  | Clarification
made or
Changes
Incorporated |
|------------|--------------------------------------------------------------------------------------------------------------------------------------------------------------------------------------------------------------------------------------------|-----------------------------------------------------------------------------------------------------------------------------------------------------------------------------------------------------------------------------------------------------------------------------------------|-----------------------------------------------------|
| 1          | 1 1. There is very little information is provided about the
compilation of earthquake catalog, the historical era
events may be discussed in the article and a small
discussion on its compilation is required to be
included. |                                                                                                                                                                                                                                                                                         | YES                                                 |
|            | The reviewer is the
manuscript no.
The catalogue
historical eart
Ambrasey and
for the catal-
remove the du                                                                                                               | hanked for this suggestion, the revised
ow incorporated the requested information.
was compiled till 2015 that also included
hquake data from Ambrasey (2000) and
I Douglas (2004). Below is the priority list
ogue homogenization and correctness to
plicate events: |                                                     |
|            | Priority                                                                                                                                                                                                                                   | Data Source                                                                                                                                                                                                                                                                             |                                                     |
|            | 1                                                                                                                                                                                                                                          | Ambrasey & Douglas (2004)                                                                                                                                                                                                                                                               |                                                     |
|            | 2                                                                                                                                                                                                                                          | Ambrasey (2000)                                                                                                                                                                                                                                                                         |                                                     |

**REVIEWER 1**

| Sr.
No. | Questions with answers                                                                                                                                                                                                                                                                                                                                                                             |      | Clarification
made or
Changes
Incorporated |
|------------|----------------------------------------------------------------------------------------------------------------------------------------------------------------------------------------------------------------------------------------------------------------------------------------------------------------------------------------------------------------------------------------------------|------|-----------------------------------------------------|
|            | 3                                                                                                                                                                                                                                                                                                                                                                                                  | ISC  |                                                     |
|            | 4                                                                                                                                                                                                                                                                                                                                                                                                  | GCMT |                                                     |
|            | 5                                                                                                                                                                                                                                                                                                                                                                                                  | NGDC |                                                     |
|            | 6                                                                                                                                                                                                                                                                                                                                                                                                  | USGS |                                                     |
|            |                                                                                                                                                                                                                                                                                                                                                                                                    | ·    |                                                     |
| 2          |  <li>2. Completeness analysis and declustering of catalog is missing, both catalog completeness and declustering should be discussed, as these seriously effect the hazard values.</li> <li>The authors fully agree with the reviewer. The requested information is provided in the revised manuscript.</li>                                                                              |      | YES                                                 |
| 3          | 3. Several important studies concerning Peshawar,
Khaliq et al. (2018); Waseem et al. (2018); Sesetayn et
al. (2018) Ahmad et al. (2019) have been missed by the
authors, they should be cited in the article, where
required.       YES         The authors thank the reviewer for this suggestion, the
revised manuscript now incorporated the mentioned
references.       YES |      | YES                                                 |
| 4          |  <li>4. The results of this study should be discussed and compared with the similar studies.</li> <li>The authors fully agree to the reviewer, the revised manuscript now compared the outcomes with relevant studies.</li>                                                                                                                                                               |      | YES                                                 |

**REVIEWER 2**

| Sr.
No. | Questions with answers                                                                                                                                                                                                                                                                                                                                                                                                                                                                                                                                                                                                                                                                                                           | Clarification
made or
Changes
Incorporated                      |
|------------|----------------------------------------------------------------------------------------------------------------------------------------------------------------------------------------------------------------------------------------------------------------------------------------------------------------------------------------------------------------------------------------------------------------------------------------------------------------------------------------------------------------------------------------------------------------------------------------------------------------------------------------------------------------------------------------------------------------------------------|--------------------------------------------------------------------------|
| 1          | 1. Please justify why you opted for areal sources
surpassing point and line sources. Or, why you did not
opt for a combination of all?                                                                                                                                                                                                                                                                                                                                                                                                                                                                                                                                                                                     | Comments are
provided herein to
clarify the
author's intention. |
|            | The reviewer is thanked for pointing to this. The available
catalogue and seismicity pattern is quite scattered and not
very specific to suggest taking line/fault sources. There
are not enough data (fault source information) to support
line sources based hazard assessment. Moreover, due to
the scattered seismicity pattern and large number of small
fault-lines, the definition of fault line becomes
challenging. Areal sources are reasonable approximation
to idealize seismic sources and used in many similar
studies. Few studies have used fault sources but that have
resulted in very high seismic hazard, which is not justified
by the history of earthquakes in Peshawar. |                                                                          |
| 2          | 2. The attenuation relationship you are using is not from
the Himalaya. Would it be possible to calibrate some
subduction GMPE with the available records from
Pakistan to obtain more realistic results?                                                                                                                                                                                                                                                                                                                                                                                                                                                                                                               |                                                                          |
|            | The authors fully agree with the reviewer. Pakistan doesn't have
specific GMPEs of its own, therefore, GMPEs from other
similar region are adopted. The selected GMPEs were
earlier tested in prediction of ground motion for selected
earthquake events. The GMPEs, which have shown
relatively better performance, were selected for hazard
analysis. This is clarified in the revised manuscript.                                                                                                                                                                                                                                                                                                           | YES                                                                      |
| 3          | 3. As you contrasted yourself, the effects of deep
earthquakes were pronounced recently in Pakistan yet
you did not include the effect. Could you please
reframe the logic tree in any way to incorporate this?                                                                                                                                                                                                                                                                                                                                                                                                                                                                                                         | Commentsareprovided herein toclarifytheauthor's intention.               |
|            | The authors fully agree with this. Deep sources were included in
the hazard assessment, however, this had little influence
on the final hazard maps. Possible reason seems to be the                                                                                                                                                                                                                                                                                                                                                                                                                                                                                                                                       |                                                                          |

| Sr.
No. | Questions with answers                                                                                                                                                                                                                                                                                                                                                                                                                                                                                                                                                                                                                                                  | Clarification
made or
Changes
Incorporated                      |
|------------|-------------------------------------------------------------------------------------------------------------------------------------------------------------------------------------------------------------------------------------------------------------------------------------------------------------------------------------------------------------------------------------------------------------------------------------------------------------------------------------------------------------------------------------------------------------------------------------------------------------------------------------------------------------------------|--------------------------------------------------------------------------|
|            | limited earthquake catalogue and the foreign GMPEs,
which were not specific to the region. Also, the GMPEs
lack to take into account the site effects common in
Peshawar valley due to deep earthquakes. We didn't find
any other alternative to manifest the deep sources effects
accurately, however, we put this as a question for others
to address.                                                                                                                                                                                                                                                                                              |                                                                          |
| 4          |  <li>4. The source zones are somehow interesting too. For instance, why zones 5, 6, and 7 have quite limited data? Could you please elucidate your zoning scheme?</li> <li>The authors fully agree to the reviewer, however, the seismic sources used for shallow earthquakes were those obtained from the Building Code of Pakistan – Seismic Provisions. The deep sources were selected in consultation with the National Center of Excellence in Geology, Peshawar. Since, deep sources are not studied before for Peshawar. This is clarified in the revised manuscript.</li>                                                                              | Comments are
provided herein to
clarify the
author's intention. |
| 5          |  <li>5. As you have prepared the hazard maps for bedrock, I request you to consider hazard maps on the surface too [if possible]. If you have some site response/amplification studies, it would be interesting and also useful for the structural earthquake engineering communities. Please comment.</li> <li>The authors fully agree to the reviewer, however, the focus of present study was to provide the base maps for hazard. Site-specific soil was not known so it was not addressed in the hazard assessment. Alternatively, the code suggests amplification factors for various soil from Type C to Type E as per NEHRP soil classification.</li>  | Comments are
provided herein to
clarify the
author's intention. |
| 6          |  <li>6. Please fix some grammatical bugs present in the manuscript.</li> <li>The authors thank the reviewer for this suggestion, the revised manuscript is re-visited for English writing improvement.</li>                                                                                                                                                                                                                                                                                                                                                                                                                                                    | YES                                                                      |

**SEISMIC HAZARD MAPS OF PESHAWAR DISTRICT FOR VARIOUS RETURN PERIODS**

**3**

[revised manuscript text omitted]

(a) Peshawar City, Qisa Khwani Bazar: Complete collapse of building roof.

(b) Peshawar City, Ganj: Sliding of overhead tank on the building roof.

---

## Author Response (AR2)

**RESPONSE TO THE EDITOR'S COMMENTS**

The authors are very grateful to the Editors and Associate Editors for the kind consideration and possible publication of our article in the Natural Hazards and Earth System Sciences. The authors would like to thank all reviewers for suggesting improvements for the manuscript. Point-wise reply/answer to each comment is provided below (comments are shown in BOLD, answers are shown in REGULAR). All suggestions have been addressed, but still if reviewers have any other point/reservation, the authors are happy to incorporate. Furthermore, the authors appreciate the editors and reviewers for the timely handling of review process.

**REVIEWER 2**

| Sr.
No. | Questions with answers                                                                                                                                                                                                                                                                                                                                                                                                                    | Clarification
made or
Changes
Incorporated |
|------------|-------------------------------------------------------------------------------------------------------------------------------------------------------------------------------------------------------------------------------------------------------------------------------------------------------------------------------------------------------------------------------------------------------------------------------------------|-----------------------------------------------------|
| 1          | 1. After, reading the revised version of the article, I consider that suggestions previously recommended are incorporated in revised article. There are some grammatical errors in article that needs to be corrected. Recently published studies about Pakistan: Updated earthquake catalog by Khan et al., 2018 and Probabilistic seismic hazard analysis using an areal source model may be cited in the article for the completeness. | YES                                                 |
|            | The reviewer is thanked for this suggestion, the revised manuscript now incorporated the requested information.                                                                                                                                                                                                                                                                                                                           |                                                     |

**SEISMIC HAZARD MAPS OF PESHAWAR DISTRICT FOR VARIOUS RETURN PERIODS**

**3**

**4 Khalid Mahmood1, Naveed Ahmad2,\*, Usman Khan1, Qaiser Iqbal1**

[revised manuscript text omitted]

Dr Naveed 3/4/20 19:45 Deleted: vicinity Dr Naveed 3/4/20 19:46 Deleted: s

Dr Naveed 3/4/20 19:46 **Deleted:** , which Dr Naveed 3/4/20 19:47 **Deleted:** e

Dr Naveed 3/4/20 19:50 Deleted:

Dr Naveed 3/4/20 19:48 Deleted: has

Dr Naveed 3/4/20 19:50 Deleted: polygon Dr Naveed 3/4/20 19:51 Deleted: used as seismic sources

1 of PGA at bedrock was calculated and plotted in GIS tool. Different ground motion attenuation

2 relationships compatible to the geology and seismicity of local environment were used to quantify model-

3 to-variability in seismic hazard of Peshawar District, Furthermore, the logic tree approach was used to

4 take into consideration the epistemic uncertainty. The GIS based seismic hazard map developed for a

return period of 475 years was compared with that given in the BCP-SP (2007). Seismic hazard maps

- 6 were prepared for various other return periods i.e. 50, 100, 250, 475 and 2500 years.
- 7

5

Dr Naveed 3/4/20 19:57

Dr Naveed 3/4/20 19:53

Dr Naveed 3/4/20 19:53

3

(a) Peshawar City, Qisa Khwani Bazar: Complete collapse of building roof.

(b) Peshawar City, Ganj: Sliding of overhead tank on the building roof.